# ENABLING FINE-TUNING OF DIRECT FEEDBACK ALIGNMENT VIA FEEDBACK-WEIGHT MATCHING

**Yunseok Lee**
UNIST
walk1009@unist.ac.kr

**Seulki Lee**
KAIST
seulki.lee@kaist.ac.kr

## ABSTRACT

Although Direct Feedback Alignment (DFA) has demonstrated potential by enabling efficient and parallel updates of weight parameters through direct propagation of the network's output error, its usage has been primarily restricted to training networks from scratch. In this paper, we introduce feedback-weight matching, a first method that enables reliable fine-tuning of fully connected neural networks using DFA. We provide an analysis showing that existing standard DFA struggles to fine-tune networks pre-trained via back-propagation. Through a thorough analysis of weight alignment (WA) and gradient alignment (GA), we demonstrate that the proposed feedback-weight matching enhances DFA's ability and stability in fine-tuning, which provides useful insights into DFA's behavior and characteristics in fine-tuning. In addition, we prove that feedback-weight matching, when combined with weight decay, not only mitigates over-fitting but also further reduces the network output error, leading to improved learning performance during DFA-based fine-tuning. Experimental results show that feedback-weight matching, for the first time, enables reliable fine-tuning across various fine-tuning tasks, compared to existing standard DFA, e.g., achieving 7.97% accuracy improvement on image classification tasks (82.67% vs. 74.70%) and 0.66 higher correlation score on NLP tasks (0.76 vs. 0.10). The code is available on an GitHub[1].

## 1 INTRODUCTION

Recently, an alternative training mechanism called *Direct Feedback Alignment (DFA)* (Nøkland, 2016) has been proposed. Based on the concept of Feedback Alignment (FA) (Lillicrap et al., 2016), DFA passes the error of the output layer directly to each layer of the network to update the weight parameters without compute-intensive back-propagation (Rumelhart et al., 1986). By using random feedback matrices, the weight gradient of each layer is independently approximated from the directly passed error, *enabling efficient training of fully connected networks* through the parallel update of multiple layers. This contrasts with the conventional back-propagation that propagates the network error sequentially from the last to the first layer.

Although Direct Feedback Alignment (DFA) (Nøkland, 2016) has shown its potential in training primarily for fully connected networks (Garg & Vempala, 2022; Launay et al., 2020), its application to fine-tuning (Devlin et al., 2018), i.e., adapting a pre-trained network to a new task, has been less studied until today despite its practical usefulness. In fact, it has been known that *fine-tuning networks with DFA is challenging* (Chu & Bacho, 2024); the performance of networks fine-tuned with DFA is generally unreliable compared to that of those fine-tuned with back-propagation (Rumelhart et al., 1986). Given that fine-tuning has become one of the practical and also effective ways of re-utilizing pre-trained networks for various downstream tasks (Church et al., 2021), *investigating how DFA can be applied to the fine-tuning mechanism both theoretically and empirically is necessary*.

Enabling fine-tuning with Direct Feedback Alignment (DFA) (Nøkland, 2016) can not only broaden DFA's usability but also introduce an alternative approach to current back-propagation-based fine-tuning (Rumelhart et al., 1986; Church et al., 2021). Currently, DFA has not yet been established as a reliable stand-alone training method that can provide comparable performance to back-propagation (Launay et al., 2019; Crafton et al., 2019). Thus, taking a wide range of well-pre-trained

---

[1] https://github.com/eai-lab/FeedbackWeightMatching

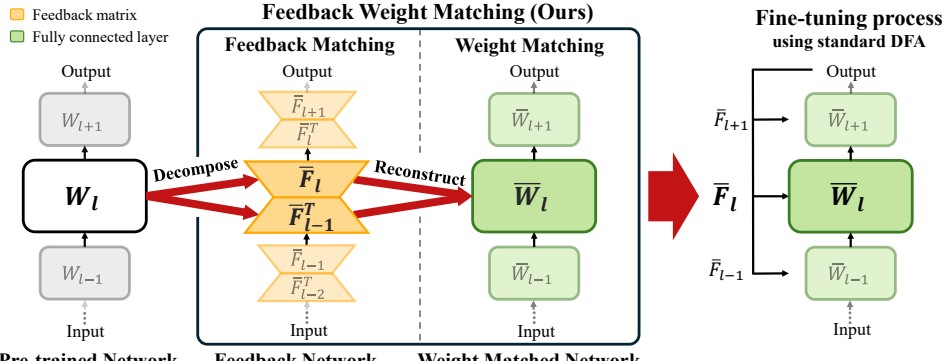

Figure 1: **An overview of the Feedback-Weight Matching process.**

models, such as Transformer-based foundation models (Kenton & Toutanova, 2019), as the starting point would be a practical strategy that can complement DFA's unstable and limited learning capabilities. Additionally, by incorporating DFA's unique advantages, such as back-propagation-free and parallel training, into the widely used fine-tuning, we can explore new possibilities for re-utilizing pre-trained models with DFA in a more agile, efficient, and biologically plausible manner, in contrast to conventional back-propagation requiring much more resources and time in model training.

In this paper, we introduce a DFA-based fine-tuning method, which investigates the feasibility of Direct Feedback Alignment (DFA) (Nøkland, 2016) for fine-tuning deep neural networks, with the aim of extending the scope of DFA to embrace various pre-trained networks. We first analyze the reasons why the existing standard DFA, which updates the pre-trained weights using random feedback matrices, does not perform well in fine-tuning. This analysis is based on the weight alignment (WA) and gradient alignment (GA) (Refinetti et al., 2021), which are two measures proposed to estimate the state and learning performance of DFA. From this analysis, we propose *feedback-weight matching* as illustrated in Fig. 1, which first reconstructs the feedback matrices by decomposing the pre-trained weights *(feedback matching)* and then re-initializes the weights based on the reconstructed feedback matrices before starting fine-tuning *(weight matching)*. Additionally, we prove that applying weight decay (Krogh & Hertz, 1991) on top of the proposed feedback-weight matching considerably improves and stabilizes the fine-tuning performance of DFA, beyond the general regularization effect on weight parameters. Together with the simple yet effective feedback-weight matching, weight decay acts as a key facilitator for fine-tuning fully connected networks with DFA. To the best of our knowledge, this work is *the first to explore the possibility of applying DFA to fine-tuning of fully connected networks* with in-depth study.

The experiments provide evaluation results consistent with our theoretical analysis; applying feedback-weight matching enables more effective and reliable fine-tuning of fully connected networks with DFA over various fine-tuning tasks, when compared to existing standard DFA (Nøkland, 2016). For instance, the image classification accuracy of fully connected networks fine-tuned with feedback-weight matching reaches 82.67%, while the standard DFA achieves 74.70%. Also, it successfully fine-tunes Transformer models (BERT) (Devlin et al., 2018) on NLP tasks, e.g., achieving 0.76 correlation score, while the standard DFA barely conducts fine-tuning at all, i.e., achieving mere 0.10 correlation score. These results demonstrate the potential for extending DFA to the widely used pre-training and fine-tuning strategy, moving beyond its limited usage in from-scratch training.

## 2 BACKGROUND AND RELATED WORK

**DFA**. It is common to train a neural network using the back-propagation algorithm (Rumelhart et al., 1986). Given a fully connected network, we denote $\mathbf{W}_l$ as the weight of $l$-th layer of the network, $\mathcal{L}(\hat{\mathbf{y}}, \mathbf{y})$ as the loss function, where $\hat{\mathbf{y}}$ is the ground-truth output, and $\mathbf{y}$ is the network output, and $\mathbf{h}_l = g(\mathbf{a}_l)$ as the output of the $l$-th layer, where $g(\cdot)$ is the activation function, and $\mathbf{a}_l = \mathbf{W}_l \mathbf{h}_{l-1}$. To update the weight parameter with the gradient descent algorithm (Ruder, 2016), the gradient of the loss $\mathcal{L}$ w.r.t. the weight $\mathbf{W}_l$ is obtained using back-propagation (BP) as:

$$\delta\mathbf{W}_l^{BP} = -\frac{\partial\mathcal{L}}{\partial\mathbf{W}_l} = -\left[\left(\mathbf{W}_{l+1}^{\top}\delta\mathbf{a}_{l+1}\right) \odot g'(\mathbf{a}_l)\right]\mathbf{h}_{l-1}^{\top}, \quad \delta\mathbf{a}_l = \partial\mathcal{L}/\partial\mathbf{a}_l \qquad (1)$$

where $\odot$ is the Hadamard product. However, back-propagation poses challenges, specifically the weight transport (Grossberg, 1987; Crick, 1989) and backward locking problems (Lillicrap et al.,

2020; Launay et al., 2019). Direct Feedback Alignment (DFA) (Nøkland, 2016) addresses the weight transport problem by employing random feedback and mitigates the backward locking problem by delivering the network's output error to each layer independently. Specifically, 1) the global error $\mathbf{e} = \hat{\mathbf{y}} - \mathbf{y}$ is transmitted to each layer, and 2) the weight $\mathbf{W_{1+1}}$ at the $l$-th layer is replaced with a random feedback matrix $\mathbf{F_1}$, leading to the following weight gradient:

$$\delta \mathbf{W}_l^{DFA} = -\frac{\partial \mathcal{L}}{\partial \mathbf{W}_l} = -\left[ (\mathbf{F}_l \mathbf{e}) \odot g'(\mathbf{a}_l) \right] \mathbf{h}_{l-1}^\top - \lambda^t \mathbf{W}_l \tag{2}$$

where $\lambda^t$ is the weight-decay hyperparameter. Eq. (2) eliminates the necessity of sequential layer-wise gradient computations of back-propagation (Rumelhart et al., 1986).

**GA and WA**. To better elucidate the dynamics of DFA (Nøkland, 2016), the concept of gradient alignment (GA) is introduced (Lillicrap et al., 2016). GA quantifies the similarity between the weight gradients obtained through DFA and those derived via back-propagation (Rumelhart et al., 1986). This is achieved by comparing the weight updates generated from the identically initialized weights by both methods. It has been hypothesized that a stronger (higher) GA corresponds to enhanced learning performance in DFA. In addition, the weight alignment (WA) (Refinetti et al., 2021) has been introduced to evaluate the relationship between the weight and the feedback matrix in DFA, suggesting that strong WA is associated with strong GA. Although GA and WA have been instrumental in analyzing the learning efficacy of DFA, prior research has not explored their utility in the context of fine-tuning. In contrast, this paper pioneers the application of GA and WA concepts to systematically investigate the fine-tuning process in DFA.

**Applying DFA to Transformers and CNNs**. Some studies (Launay et al., 2020) explore the applicability of DFA (Nøkland, 2016) to various fully connected networks, including NeRF (Mildenhall et al., 2021; Sitzmann et al., 2019), recommender systems (Guo et al., 2017), and NLP (Vaswani, 2017; Merity et al., 2016). While they show that DFA can train a range of deep architectures, they also reveal a significant performance gap between DFA and back-propagation (Rumelhart et al., 1986), particularly in Transformer models (Vaswani, 2017). When applied to models not based on fully connected networks, such as CNNs, the performance gap between DFA and back-propagation is even more pronounced. For instance, VGG-16 (Simonyan & Zisserman, 2014) on CIFAR-100 (Krizhevsky et al., 2009) trained with DFA achieves 1% top-1 accuracy (Launay et al., 2019), while back-propagation achieves 60%. Similarly, in ImageNet (Deng et al., 2009), it is 6.2% vs. 53% (Crafton et al., 2019). Given that applying DFA to from-scratch training scenarios 1) consistently underperforms relative to back-propagation, 2) takes a much longer training time than fine-tuning, and 3) is limited to a narrower range of architectures, we argue that utilizing fine-tuning for DFA would be a more effective, efficient, practical, and expedient approach. Thus, in this study, we investigate the potential of employing DFA in fine-tuning, which is conducive to the widely-used pre-train-and-fine-tune strategy (Devlin et al., 2018).

**Applying DFA to back-propagation weights**. As described above, in CNNs, DFA encounters challenges in effectively learning the necessary spatial information (Crafton et al., 2019). Similarly, in fully connected networks, DFA is known to produce feature representation that deviate from those learned via back-propagation (Nøkland, 2016). Moreover, although stable training can be achieved when transitioning from weights learned through DFA to back-propagation, the reverse is not true; switching from back-propagation to DFA results in unstable training, and DFA fails to fully recover its performance even after large train epochs (Chu & Bacho, 2024). These imply inherent difficulties in fine-tuning with DFA using pre-trained weights.

**DFA with weight decay**. In the prior study Song et al. (2021), it is analyzed that weight decay (Krogh & Hertz, 1991) can reduce the output error in fully connected networks when used with Feedback Alignment (FA) (Lillicrap et al., 2016). Nevertheless, the analysis predominantly focuses on training of networks from scratch using FA, rather than on the fine-tuning process with DFA. This work, for the first time, examines the impact of weight decay in the context of fine-tuning with DFA. Our findings show that weight decay can be beneficial in fine-tuning with DFA, as it reduces the network output error, enhancing learning performance.

## 3 FEEDBACK-WEIGHT MATCHING

We first discuss why the existing standard DFA (Nøkland, 2016) does not behave stably in fine-tuning, based on weight alignment (WA) and gradient alignment (GA) (Refinetti et al., 2021). Then,

we introduce feedback-weight matching, which enables effective and reliable fine-tuning of DFA with two phases: 1) feedback matching and 2) weight matching. In the first phase, feedback matrices are reconstructed from the pre-trained weights. In the second phase, network weights are re-initialized to align with the reconstructed feedback matrices. Then, fine-tuning is performed using the standard DFA method.

## 3.1 WHY DOES DFA PERFORM UNRELIABLY IN FINE-TUNING?

**Definition 3.1. (Weak Weight Alignment)** *Given a $L$-layer fully connected linear network updated (trained) with DFA (Nøkland, 2016), the weight parameter of the $l$-th layer at the $t$-th training step, which is denoted as $\mathbf{W}^t_{1 \leq l \leq L}$, becomes as follows (Refinetti et al., 2021):*

$$\mathbf{W}^t_1 = \mathbf{F_1}\mathbf{A}^t_1, \ \mathbf{W}^t_{1<l<L} = \mathbf{F}_l\mathbf{A}^t_l\mathbf{F}^\top_{l-1}, \ and \ \mathbf{W}^t_L = \mathbf{A}^t_L\mathbf{F}^\top_{L-1}, \tag{3}$$

$$where \ \mathbf{A}^t_1 = -\eta \sum_{t'=0}^{t-1} \mathbf{e}^{t'}(\mathbf{x}^{t'})^\top, and \ \mathbf{A}^t_{l \geq 2} = \eta^2 \sum_{t'=0}^{t-1}\sum_{t''=0}^{t'-1} (\mathbf{B}^{t'}_l\mathbf{x}^{t'})(\mathbf{B}^{t''}_l\mathbf{x}^{t''})\mathbf{e}^{t'}(\mathbf{e}^{t''})^\top$$

Here, $\mathbf{F}_l$ is the feedback matrix of the $l$-th layer, $\mathbf{A}^t_1$ and $\mathbf{A}^t_{l \geq 2}$ are the alignment matrices, and $\mathbf{B}_l = \mathbf{A}_{l-2}\cdots\mathbf{A}_0 \in \mathbb{R}^{n_L \times n_L}$ is defined recursively using the feedback matrices only, with $\mathbf{A}_0 = \mathbf{I}$ (Refinetti et al., 2021). Eq. (3) is referred to as *weak weight alignment (WA)* (Refinetti et al., 2021), representing the state where no particular relationship exists between $\mathbf{W}^t_{1<l<L}$ and $\mathbf{F}_l\mathbf{F}^\top_{l-1}$ and between $\mathbf{W}^t_L$ and $\mathbf{F}^\top_{L-1}$. At the early stage of DFA training, weak WA is naturally induced since $\mathbf{A}^t_{l \geq 2}$ in Eq. (3) starts with arbitrary values. However, as the training proceeds, $\mathbf{A}^t_{l \geq 2}$ becomes proportional to the identity matrix (Refinetti et al., 2021), i.e., $\mathbf{A}^t_{l \geq 2} \propto \mathbf{I}$, leading to another state called *strong weight alignment (WA)*, which is defined as follows.

**Definition 3.2. (Strong Weight Alignment)** *If $\mathbf{A}^t_{l \geq 2} \propto \mathbf{I}$, Eq. (3) becomes the state called* strong weight alignment (WA)*, which is defined as follows.*

$$\mathbf{W}^t_{1<l<L} \propto \mathbf{F}_l\mathbf{F}^\top_{l-1}, \ \mathbf{W}^t_L \propto \mathbf{F}^\top_{L-1} \tag{4}$$

It is known that the strong WA in Eq. (4), given $\mathbf{F}^\top_l\mathbf{F}_l \equiv \mathbf{I}$, implies *strong gradient alignment (GA)* (Refinetti et al., 2021) defined in Eq. (9), causing the gradient direction of the DFA weight, $\mathbf{W}^t_{1<l\leq L}$, aligned to that of back-propagation (Rumelhart et al., 1986). Hence, strong WA leads the learning trajectory of DFA to be comparable to that of back-propagation with strong GA.

However, if the pre-trained weights are fine-tuned via existing standard DFA using arbitrary random feedback matrix $\mathbf{F}_l$, it becomes difficult to achieve strong WA in Eq. (4), as shown below, likely to result in sub-optimal fine-tuning performance by inducing weak GA from weak WA.

**Proposition 3.3.** *If the pre-trained weight, $\mathbf{W}^0_l$, is updated via DFA with arbitrary random feedback matrices $\mathbf{F}_l$, the strong WA condition in Eq. (4) is unlikely to be satisfied as:*

$$\mathbf{W}^t_{1<l<L} \not\propto \mathbf{F}_l\mathbf{F}^\top_{l-1}, \ \mathbf{W}^t_L \not\propto \mathbf{F}^\top_{L-1} \tag{5}$$

*where $\mathbf{W}^t_l$ denotes the weight after $t$ steps of training, starting from the pre-trained weight $\mathbf{W}^0_l$.*

*Proof.* The proof is provided in Sec. A. □

Thus, Eq. (5) implies that simply applying the standard DFA to the pre-trained weight parameters is less likely to induce the strong WA condition in Eq. (4).

## 3.2 INDUCING STRONG WEIGHT ALIGNMENT

To enable fine-tuning with DFA by deriving strong GA from strong WA defined in Eq. (4), we propose the *feedback-weight matching* method, which induces both strong WA and GA as follows.

**Definition 3.4. (Feedback Matching)** *From the pre-trained weight $\mathbf{W}^0_l$, we set the feedback matrix $\bar{\mathbf{F}}_l$ such that:*

$$\bar{\mathbf{F}}_l\bar{\mathbf{F}}^\top_{l-1} \approx \mathbf{W}^0_{1<l<L} \ and \ \bar{\mathbf{F}}^\top_{L-1} \equiv \mathbf{W}^0_L. \tag{6}$$

Eq. (6) requires decomposing the pre-trained weight $\mathbf{W}^0_{1<l<L}$ into $\bar{\mathbf{F}}_l$ and $\bar{\mathbf{F}}^\top_{l-1}$. It can be achieved either through traditional methods, such as SVD (Singular Value Decomposition) (Klema & Laub, 1980), or alternatively, by optimizing Eq. (23), as in Sec. B. Once the feedback matrix $\bar{\mathbf{F}}_l$ is reconstructed as Eq. (6), we proceed to the *weight matching* process to induce strong WA, as follows.

**Definition 3.5.** (**Weight Matching**) *Given the reconstructed $\bar{\mathbf{F}}_l$ derived by feedback matching, as in Eq.* (6), *we re-initialize the pre-trained weight $\mathbf{W}_l^0$ into $\bar{\mathbf{W}}_l^0$ so that it matches $\bar{\mathbf{F}}_l$ such that:*

$$\bar{\mathbf{W}}_{1<l<L}^0 \equiv \bar{\mathbf{F}}_l \bar{\mathbf{F}}_{l-1}^\top \ and \ \bar{\mathbf{W}}_L^0 \equiv \bar{\mathbf{F}}_{L-1}^\top. \tag{7}$$

The following proposition shows that Eq. (6) and (7) together lead to strong WA condition in Eq. (4).

**Proposition 3.6.** *If the re-initialized weight $\bar{\mathbf{W}}_l^0$ in Eq.* (7) *is updated using DFA with the feedback matrix $\bar{\mathbf{F}}_l$ derived by Eq.* (6), *the strong WA condition in Eq.* (4) *is induced as:*

$$\bar{\mathbf{W}}_{1<l<L}^t \propto \bar{\mathbf{F}}_l \bar{\mathbf{F}}_{l-1}^\top, \ \bar{\mathbf{W}}_L^t \propto \bar{\mathbf{F}}_{L-1}^\top \tag{8}$$

*where $\bar{\mathbf{W}}_l^t$ is the weight at step $t$, initialized from $\bar{\mathbf{W}}_l^0$.*

*Proof.* The proof is provided in Sec. A. □

Thus, Eq. (8) indicates that applying feedback-weight matching to the weight updated from the re-initialized weight induces the strong WA condition in Eq. (4), in contrast to standard DFA (Eq. (5)). Subsequently, strong WA, achieved through Eq. (6) and Eq. (7), leads to strong GA (Refinetti et al., 2021). By matching the feedback matrix to the pre-trained weights, as in Eq. (6), it becomes possible to preserve the knowledge embedded in the pre-trained weights. Additionally, by re-initializing the pre-trained weights from the matched feedback matrices, as in Eq. (7), it becomes possible to facilitate the attainment of strong WA through DFA in fine-tuning.

### 3.3 Inducing strong gradient alignment

While the previous section (Sec. 3.2) shows that the proposed feedback-weight matching in Eq. (6) and (7) promotes strong weight alignment (WA), naturally leading to strong gradient alignment (GA), we now show that feedback-weight matching also directly induces strong GA. We begin by formally defining gradient alignment (GA) as follows.

**Definition 3.7.** (**Gradient Alignment**) *The gradient alignment (GA) is defined as the cosine similarity between the weight gradient obtained using DFA (Nøkland, 2016), denoted as $\mathbf{G}_{DFA}$, and the weight gradient of back-propagation (Rumelhart et al., 1986), denoted as $\mathbf{G}_{BP}$, which is given by:*

$$\cos \angle (\mathbf{G}_{DFA}, \mathbf{G}_{BP}) = \mathbf{G}_{DFA} \cdot \mathbf{G}_{BP} / \|\mathbf{G}_{DFA}\| \|\mathbf{G}_{BP}\|. \tag{9}$$

We show that feedback-weight matching, i.e., Eq. (6) and (7), also directly induce strong GA when fine-tuning the first layer of the two-layer fully connected linear network.

**Proposition 3.8.** *Feedback-weight matching given in Eq.* (6) *and* (7) *induces strong GA, i.e., a higher GA, in the first layer of a fully connected linear network, as follows:*

$$\cos_{FWM} \angle (\mathbf{F}_1, \mathbf{W}_2^t) \geq \cos_{DFA} \angle (\mathbf{F}_1, \mathbf{W}_2^t) \tag{10}$$

*where $\cos_{FWM} \angle (\mathbf{F}_1, \mathbf{W}_2^t)$ refers to GA in the first layer using feedback-weight matching, while $\cos_{DFA} \angle (\mathbf{F}_1, \mathbf{W}_2^t)$ is GA in the first layer using standard DFA without feedback-weight matching.*
*Proof.* The proof is provided in Sec. A. □

## 4 Weight decay

Similar to conventional trains using back-propagation (Nøkland, 2016), weight decay (Krogh & Hertz, 1991) has been shown to mitigate over-fitting of DFA, though its effect in fine-tuning has not been studied. We discuss how the proposed feedback-weight matching helps weight decay to reduce the network error (i.e., improving learning performance) in fine-tuning when applied to DFA.

**Lemma 4.1.** *Given the re-initialized weight $\bar{\mathbf{W}}_{1<l\leq L}^0$ in Eq.* (7) *and the pre-trained weight $\mathbf{W}_{1<l\leq L}^0$, the following two terms, $r_{1<l<L}$ and $r_L$, become non-negative with high probability.*

$$r_{1<l<L} = \|\mathbf{W}_l^t - \mathbf{W}_l^0\| - \|\mathbf{W}_l^t - \bar{\mathbf{W}}_l^0\| = \|\bar{\mathbf{F}}_l \bar{\mathbf{F}}_{l-1}^\top - \mathbf{W}_l^0\| - |c_l^t - 1| \|\bar{\mathbf{F}}_l \bar{\mathbf{F}}_{l-1}^\top\| \geq 0, \tag{11}$$

$$r_L = \|\mathbf{W}_L^t - \mathbf{W}_L^0\| - \|\mathbf{W}_L^t - \bar{\mathbf{W}}_L^0\| = \|\bar{\mathbf{F}}_{L-1}^\top - \mathbf{W}_L^0\| - |c_L^t - 1| \|\bar{\mathbf{F}}_{L-1}^\top\| \geq 0, \tag{12}$$

*implying $\|\mathbf{W}_l^t - \mathbf{W}_l^0\| \geq \|\mathbf{W}_l^t - \bar{\mathbf{W}}_l^0\|$ for all $1 < l \leq L$.*
*Proof.* The proof is provided in Sec. A. □

Based on Lem. 4.1, we show that feedback-weight matching reduces the network output error $\mathbf{e}^{t+1}$ over the step $t$ when combined with weight decay (Krogh & Hertz, 1991).

**Proposition 4.2.** *Let $\mathbf{e}^t$ denote the output error of a two-layer fully connected non-linear network (i.e., L=2) at the t-th training step, $\eta$ is the learning rate, $\gamma \leq \lambda_{min}(\bar{\mathbf{G}})$ is a positive constant, where $\bar{\mathbf{G}} = \mathbb{E}_{w \sim \mathcal{N}(0, I_p)} \psi(\mathbf{w}^\top \mathbf{x}_i) \psi(\mathbf{w}^\top \mathbf{x}_j)$ with the number of neuron as p and a non-linear function $\psi(\cdot)$, $\lambda^t$ is the weight-decay hyperparameter at the step t, and $\mathbf{y}$ is the output of the network. By applying feedback-weight matching in Eq. (6) and (7), the following holds:*

$$\|\mathbf{e}^{t+1}\| \leq \left(1 - \frac{\eta\gamma}{4} - \eta\lambda^t\right)\|\mathbf{e}^t\| + \lambda^t\|\mathbf{y}\| - \alpha_2 r_2 \tag{13}$$

*for all $t \geq 0$ and a constant $\alpha_2$, with $r_2$ defined in Eq. (11).*

*Proof.* The proof is provided in Sec. A. □

**Conjecture 4.3.** *Given an L-layer fully connected non-linear network, suppose that the right-hand side of the inequality in Eq. (13), i.e., $\left(1 - \frac{\eta\gamma}{4} - \eta\lambda^t\right)\|\mathbf{e}^t\| + \lambda^t\|\mathbf{y}\|$, contains $\|\mathbf{W}_l^t - \mathbf{W}_l^0\|$ as linear components for some $1 < l \leq L$. Then, based on Prop. 4.2 and Lem. 4.1, it is conjectured that Eq. (13) can be generalized into:*

$$\|\mathbf{e}^{t+1}\| \leq \left(1 - \frac{\eta\gamma}{4} - \eta\lambda^t\right)\|\mathbf{e}^t\| + \lambda^t\|\mathbf{y}\| - \sum_{l=2}^{L} \alpha_l r_l \tag{14}$$

*with constants $\alpha_l$, and $r_l$ defined in Eq. (11) and (12) for some $1 < l \leq L$ and all $t \geq 0$.*

From Eq. (13), and subsequently Eq. (14), it can be seen that the proposed feedback-weight matching preserves the weight decay effect by decreasing the network error $\|\mathbf{e}^{t+1}\|$ by the quantity $\eta\lambda^t\|\mathbf{e}^t\| - \lambda^t\|\mathbf{y}\|$. It is achieved by $\sum_{l=2}^{L} \alpha_l r_l$, which effectively counteracts the adverse impact of weight decay, namely, the increase in error $\|\mathbf{e}^{t+1}\|$ when $\eta\|\mathbf{e}^t\| \leq \|\mathbf{y}\|$, if $\sum_{l=2}^{L} \alpha_l r_l \geq \lambda^t\|\mathbf{y}\| - \eta\lambda^t\|\mathbf{e}^t\|$. Due to the dependence of the term $\|\mathbf{W}_l^t - \bar{\mathbf{W}}_l^0\|$ on the reconstructed weights, deriving a theoretical analysis is challenging. Nonetheless, the effect of weight decay on error reduction is empirically validated through our controlled experiments, as detailed in the experimental section.

## 5 EXPERIMENT

We evaluate the proposed feedback-weight matching on a wide range of fine-tuning tasks. To the best of our knowledge, this is the first extensive experiments to investigate fine-tuning of DFA across diverse tasks and models, which is enabled by the proposed feedback-weight matching.

First, feed-weight matching is applied to image classification tasks using two fully connected networks with 4 and 6 layers, respectively. These networks are pre-trained with CIFAR-100 (Krizhevsky et al., 2009) and TinyImageNet (Le & Yang, 2015) using back-propagation (Rumelhart et al., 1986), and then fine-tuned on CIFAR-10 (Krizhevsky et al., 2009), SVHN (Netzer et al., 2011), and STL-10 (Coates et al., 2011) through DFA applying feedback-weight matching. Next, we apply it to NLP tasks using Transformers, i.e., BERT-Tiny and Small (Kenton & Toutanova, 2019; Turc et al., 2019), pre-trained on BookCorpus (Zhu et al., 2015) & Wikipedia, and then fine-tuned with GLUE tasks (Wang, 2018). Lastly, we apply feedback-weight matching to Vision Transformer (ViT) models, i.e., ViT-Tiny and ViT-Small (Wu et al., 2022). Each model utilizes pre-trained weights learned from ImageNet (Deng et al., 2009) and is fine-tuned on CIFAR-10 (Krizhevsky et al., 2009), STL-10 (Coates et al., 2011), and ImageNette (Howard, 2019). It is important to note that even standard DFA has rarely been applied to Transformer models for from-scratch training due to its inherent challenges and difficulties (Launay et al., 2020). Our experiment is the first attempt to apply DFA fine-tuning to Transformers (i.e., BERT, ViT), which is considered more challenging than from-scratch DFA training. The detailed experimental setups are provided in Sec. I.

### 5.1 FINE-TUNING PERFORMANCE

Tab. 1 summarizes the fine-tuning performance on image classification tasks (i.e., test accuracy) of feedback-weight matching compared against the standard DFA fine-tuning that does not apply feedback-weight matching. As shown in the table, the proposed feedback-weight matching enables reliable fine-tuning for various network architectures and tasks, which consistently outperforms standard DFA with an average of 2.16% accuracy gap. For instance, feedback-weight matching achieves 82.67% accuracy when fine-tuning the 6-layer network from CIFAR-100 to SVHN, which is 7.97% higher than standard DFA that achieves 74.70%. It also indicates that the proposed feedback-weight matching maintains more robust performance over network depths, whereas the performance of standard DFA deteriorates with deeper networks. For instance, in the case of fine-tuning from

Table 1: **Image classification tasks.** The fine-tuning performance of feedback-weight matching (DFA$_{ours}$) on the 4- and 6-layer fully connected networks, compared with standard DFA fine-tuning (DFA$_{fine}$), and from-scratch-training of DFA (DFA$_{scratch}$). The pre-trained weights are obtained through back-propagation. The bold indicates the best performance in DFA fine-tuning.

| Target Data | 4-layer | | | | | 6-layer | | | | |
|---|---|---|---|---|---|---|---|---|---|---|
| | Scratch | CIFAR-100 | | TinyImageNet | | Scratch | CIFAR-100 | | TinyImageNet | |
| | DFA$_{scratch}$ | DFA$_{fine}$ | DFA$_{ours}$ | DFA$_{fine}$ | DFA$_{ours}$ | DFA$_{scratch}$ | DFA$_{fine}$ | DFA$_{ours}$ | DFA$_{fine}$ | DFA$_{ours}$ |
| CIFAR-10 | 52.78 | 53.79 | **55.38** | **56.75** | 55.51 | 51.94 | 53.04 | **55.39** | 51.08 | **55.54** |
| SVHN | 82.93 | 79.55 | **82.87** | 80.31 | **83.16** | 81.89 | 74.70 | **82.67** | 76.03 | **81.39** |
| STL-10 | 42.20 | 44.83 | **45.30** | **50.62** | 45.61 | 40.48 | 43.42 | **45.28** | 43.33 | **45.21** |

CIFAR-100 to SVHN, the accuracy drop between 4-layer and 6-layer networks is only 0.20% with feedback-weight matching, which is 24x smaller than the case not applying it (i.e., 4.85% drop).

Tab. 2 presents the evaluation results of feedback-weight matching applied to BERT-Tiny and BERT-Small, fine-tuned for NLP tasks, using the same experimental setup in image classification tasks (Tab. 1). Similar to image classification tasks, feedback-weight matching enables DFA to fine-tune BERT for various tasks of the GLUE dataset in a more robust manner compared to standard DFA. In particular, substantial performance gains are observed on datasets with limited samples, where fine-tuning primarily depends on pre-trained weights. For example, on CoLA, feedback-weight matching achieves a Matthews correlation of 0.53 in BERT-Small, compared to 0.06 with standard DFA. Similarly, on STSB, BERT-Small achieves a Pearson correlation of 0.76 with feedback-weight matching, while standard DFA yields only 0.10, demonstrating a significant gap in both learning performance and reliability. In the worst case, standard DFA fails to learn from the fine-tuning data entirely, achieving 0.00 Matthews correlation for CoLA with BERT-Tiny, whereas feedback-weight matching achieves 0.29. Unlike architectures with stacked fully connected layers, transformers incorporate attention mechanisms, which interfere with alignment. While alignment cannot be directly adjusted for projection layers such as key, query, and value, aligning the subsequent dense layers alone significantly improves weight alignment, gradient alignment, and overall performance. Experiments related to this are presented in Sec. E

Table 2: **NLP tasks.** The fine-tuning performance of feedback-weight matching (DFA$_{ours}$) on Transformer architectures (i.e., BERT-Tiny and BERT-Small), compared with standard DFA fine-tuning (DFA$_{fine}$). The pre-trained weights are obtained via back-propagation (BP). For reference, we also present the from-scratch-training results of DFA (DFA$_{scratch}$). The bold indicates the best performance in DFA fine-tuning.

| Model | Training | CoLA (mat) | SST-2 (acc) | MRPC (acc) | QQP (acc) | MNLI (acc) | QNLI (acc) | STSB (pear) | RTE (acc) | WNLI (acc) |
|---|---|---|---|---|---|---|---|---|---|---|
| BERT Tiny | DFA$_{scratch}$ | 0.00 | 95.2 | 67.4 | 81.2 | 59.2 | 84.2 | -0.11 | 50.2 | 50.0 |
| | DFA$_{fine}$ | 0.00 | 92.4 | 67.4 | 80.6 | 60.0 | 80.2 | -0.17 | 51.2 | 51.0 |
| | DFA$_{ours}$ | **0.29** | **95.9** | **69.7** | **82.3** | **60.2** | **84.3** | **0.36** | **55.5** | **52.6** |
| BERT Small | DFA$_{scratch}$ | 0.19 | 96.5 | 75.2 | 86.7 | 67.4 | 80.9 | 0.05 | 60.0 | 50.3 |
| | DFA$_{fine}$ | 0.06 | 95.6 | 70.9 | 86.0 | **67.0** | 85.3 | 0.10 | 59.0 | 49.3 |
| | DFA$_{ours}$ | **0.53** | **97.3** | **92.5** | **86.9** | 65.8 | **87.2** | **0.76** | **59.0** | **51.0** |

Tab. 3 shows fine-tuning results for image classification tasks using Vision Transformers (ViTs). Consistent with previous evaluations, the experiment results confirm that feedback-weight matching is effective not only for simple fully connected networks but also for complex Transformer architectures, e.g., the classification accuracy improves from 0.210 to 0.319 for ViT-Small on ImageNette.

Table 3: **Image classification tasks with Vision Transformers.** The fine-tuning performance of feedback-weight matching (DFA$_{ours}$) on ViT-Small and ViT-Tiny compared with standard DFA fine-tuning (DFA$_{fine}$) and DFA training from scratch (DFA$_{scratch}$). Bold indicates the best performance.

| Target Data | ViT-Tiny | | | ViT-Small | | |
|---|---|---|---|---|---|---|
| | DFA$_{scratch}$ | DFA$_{fine}$ | DFA$_{ours}$ | DFA$_{scratch}$ | DFA$_{fine}$ | DFA$_{ours}$ |
| CIFAR-10 | 0.281 | 0.332 | **0.397** | 0.322 | 0.378 | **0.392** |
| STL-10 | 0.197 | 0.164 | **0.247** | 0.221 | 0.111 | **0.247** |
| ImageNette | 0.168 | 0.209 | **0.294** | 0.230 | 0.210 | **0.319** |

## 5.2 WEIGHT ALIGNMENT AND GRADIENT ALIGNMENT

Fig. 2a and 2b plot the weight alignment (WA) and the gradient alignment (GA), along with the train and test accuracy, for some fine-tuning setups. As shown in the figures, the proposed feedback-

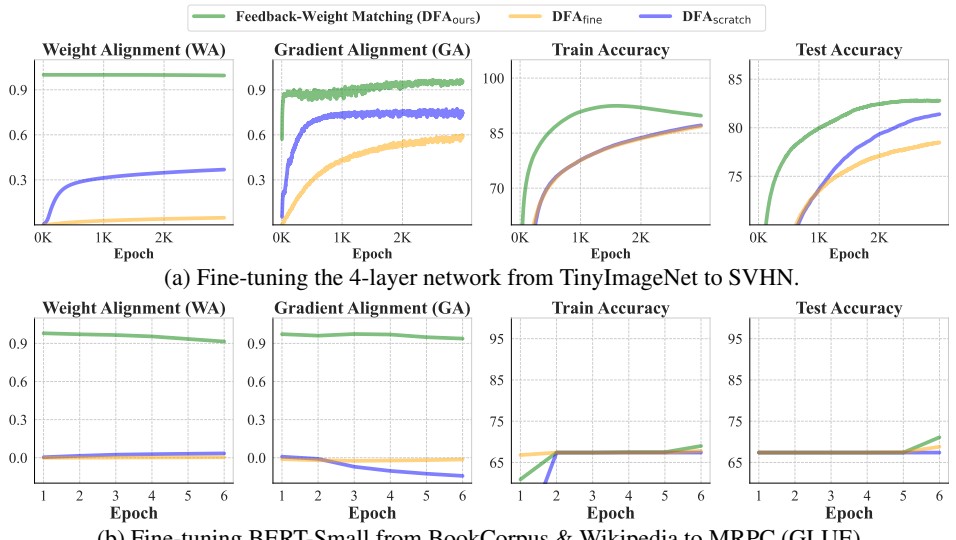

(a) Fine-tuning the 4-layer network from TinyImageNet to SVHN.

(b) Fine-tuning BERT-Small from BookCorpus & Wikipedia to MRPC (GLUE).

Figure 2: **WA, GA, train accuracy, and test accuracy.** The green graph denotes DFA fine-tuning with feedback-weight matching (ours), yellow denotes DFA fine-tuning without feedback-weight matching, blue is DFA trained from scratch.

weight matching (green) induces strong weight alignment (WA) from the outset, subsequently strong gradient alignment (GA) as analyzed in Sec. 3.2 and 3.3, leading to both enhanced train and test accuracy across all experiments with faster and stable convergence. In contrast, standard DFA (yellow), not applying feedback-weight matching, achieves significantly lower WA and GA. While they gradually increase over fine-tuning epochs in some cases, the initially low WA and GA impede effective fine-tuning. As a result, the train and test accuracy of standard DFA do not improve substantially from the pre-trained weight parameters, especially for BERT-Small. This suggests that standard DFA struggles to adapt to the target dataset for fine-tuning, likely due to the mismatch between its random feedback matrices and the pre-trained weights. In other words, it overly relies on pre-trained weights in the hope that they will fit and perform well on new target fine-tuning data. These results indicate that the improvements in fine-tuning performance are attributed to the proposed feedback-weight matching. This is well exemplified in Fig. 2b, where the train and test accuracies of DFA$_{fine}$ show minimal or no improvement from the pre-trained weights, whereas those of DFA$_{fine}$ exhibit notable increases of approximately up to 0.27 from the pre-trained weights. In addition, Transformer models achieve lower WA and GA than fully connected networks when trained with DFA from scratch. This is potentially due to the complexity of the attention operation (Sec. E). When the proposed feedback-weight matching is applied to Transformer models, it induces WA and GA, yielding better performance compared to fully connected networks.

## 5.3 ABLATION STUDY: FEEDBACK MATCHING, WEIGHT MATCHING, AND WEIGHT DECAY

Tab. 4 presents the impact of feedback matching, weight matching, and weight decay on fine-tuning with DFA. To assess their effectiveness, we remove each of them in isolation. Removing feedback matching results in a marginal performance decline, such as a reduction from 55.54% to 55.03% when the 6-layer network is fine-tuned from TinyImageNet to CIFAR-10. This marginal drop occurs because bypassing feedback matching applies random feedback matrices to the re-initialized weights that are amenable to arbitrary random feedback matrices, resulting in a reasonable level of WA and GA. In contrast, omitting weight matching leads to a relatively bigger performance drop, e.g., classification accuracy decreases from 83.16% to 79.77% when fine-tuning the 4-layer network from TinyImageNet to SVHN. Similarly, the correlation score drops from 0.76 to -0.06 when fine-tuning BERT-Small to STSB as shown in Tab. 6 (Sec. D). It is presumed that excluding weight matching causes the pre-trained weights obtained by back-propagation, not by DFA, to be fine-tuned with mismatched feedback matrices, thus resulting in weak WA and GA.

When weight decay is not applied, fine-tuning of feedback-weight matching performance also exhibits some declines, e.g., classification accuracy decreases from 55.38% to 48.82% when fine-tuning the 4-layer network from CIFAR-100 to CIFAR-10. It should be noted that weight decay appears to have minimal impact on fine-tuning of standard DFA when feedback-weight matching is

Table 4: **Ablation experiment.** The fine-tuning performance when removing each component of feedback-weight matching: weight matching (DFA$_{weight*}$), feedback matching (DFA$_{feed*}$), and weight decay (DFA$_{decay*}$). 'DFA$_{ours}$' denotes applying all of them.

| Model | Target Data | Source Data | | | | | | | |
|---|---|---|---|---|---|---|---|---|---|
| | | CIFAR-100 | | | | TinyImageNet | | | |
| | | DFA$_{weight*}$ | DFA$_{feed*}$ | DFA$_{decay*}$ | DFA$_{ours}$ | DFA$_{weight*}$ | DFA$_{feed*}$ | DFA$_{decay*}$ | DFA$_{ours}$ |
| 4 layers | CIFAR-10 | 53.92 | 55.23 | 48.82 | **55.38** | 53.73 | 55.05 | 48.66 | **55.51** |
| | SVHN | 80.65 | 81.34 | 77.99 | **82.87** | 79.77 | 83.13 | 77.63 | **83.16** |
| | STL-10 | 44.25 | 45.20 | 40.00 | **45.30** | 44.05 | 45.42 | 40.47 | **45.61** |
| 6 layers | CIFAR-10 | 53.47 | 55.03 | 46.21 | **55.39** | 53.50 | 55.03 | 45.77 | **55.54** |
| | SVHN | 79.70 | 82.53 | 76.71 | **82.67** | 79.77 | 82.76 | 76.76 | **82.72** |
| | STL-10 | 43.86 | **45.42** | 39.17 | 45.28 | 43.78 | **45.43** | 40.23 | 45.21 |

not applied; the classification accuracy even increases, such as from 54.38% to 56.75% when fine-tuning the 4-layer network from TinyImageNet to CIFAR-10. This demonstrates the synergistic effect of feedback-weight matching and weight decay, i.e., reducing network error as in Sec. 4.

## 5.4 FEEDBACK-WEIGHT MATCHING AND WEIGHT DECAY

To evaluate the impact of feedback-weight matching on weight decay, we measure the fine-tuning performance with weight decay, with and without applying feedback-weight matching, which is shown in Tab. 5. The results indicate that weight decay enhances fine-tuning accuracy (reducing network output error) when used with feedback-weight matching, with an average improvement of 8.35%. This demonstrates that feedback-weight matching facilitates weight decay in reducing network output error, thus improving fine-tuning accuracy, as provided in Eq. (14). In contrast, weight decay is less likely to improve fine-tuning performance without feedback-weight matching. In fact, when applied to the standard DFA (not applying feedback-weight matching), weight decay results in fine-tuning accuracy with minimal variation, providing similar accuracy.

Table 5: **Feedback-Weight Matching and weight decay.** 'DFA$_{fine}$' applies weight decay without feedback-weight matching, compared with 'DFA$_{ours}$' applying both weight decay and feedback-weight matching. The following tables show the results for image classification and NLP tasks.

(a) Fine-tuning image classification tasks (fully connected networks)

| Target Data | 4 layers | | | | 6 layers | | | |
|---|---|---|---|---|---|---|---|---|
| | CIFAR-100 | | TinyImageNet | | CIFAR-100 | | TinyImageNet | |
| | DFA$_{fine}$ | DFA$_{ours}$ | DFA$_{fine}$ | DFA$_{ours}$ | DFA$_{fine}$ | DFA$_{ours}$ | DFA$_{fine}$ | DFA$_{ours}$ |
| CIFAR-10 | 54.39 | **55.38** | 54.38 | **55.51** | 54.08 | **55.39** | 53.50 | **55.54** |
| SVHN | 80.77 | **82.87** | 80.74 | **83.16** | 78.73 | **82.67** | 79.57 | **82.72** |
| STL-10 | 45.00 | **45.30** | **50.40** | 45.61 | 43.56 | **45.28** | 45.28 | 45.21 |

(b) Fine-tuning NLP tasks (BERT)

| Model | Training | CoLA (mat-cor) | SST-2 (acc) | MRPC (acc) | QQP (acc) | MNLI (acc) | QNLI (acc) | STSB (pearson) | RTE (acc) | WNLI (acc) |
|---|---|---|---|---|---|---|---|---|---|---|
| BERT-Tiny | DFA$_{fine}$ | 0.00 | 92.4 | 67.4 | 80.6 | 60.0 | 80.2 | -0.17 | 51.2 | 51.0 |
| | DFA$_{ours}$ | **0.29** | **95.9** | **69.7** | **82.3** | **60.2** | **84.3** | **0.36** | **55.5** | **52.6** |
| BERT-Small | DFA$_{fine}$ | 0.06 | 95.6 | 70.9 | 86.0 | **67.0** | 85.3 | 0.10 | 59.0 | 49.3 |
| | DFA$_{ours}$ | **0.53** | **97.3** | **92.5** | **86.9** | 65.8 | **87.2** | **0.76** | 59.0 | **51.0** |

Fig. 3 plots the weight alignment (WA), gradient alignment (GA), training accuracy, and test accuracy across varying strengths of weight decay during the fine-tuning of 4-layer network from CIFAR-100 to CIFAR-10. Feedback-weight matching ensures strong WA and GA as discussed in Sec. 3.2 and 3.3 from the beginning, which helps mitigate alignment degradation, while exhibiting varying behaviors depending on different levels of weight decay. In the absence of weight decay (black curve), GA declines and exhibits significant oscillations, ultimately causing a decrease in test accuracy. Conversely, when a strong weight decay is applied (blue curve), both WA and GA decrease sharply, followed by substantial reductions in both training and test accuracy. This supports Conj. 4.3 that applying an appropriate level of weight decay can mitigate its adverse effect (the increase in error), thus leading to an overall error reduction. These observations suggest that a proper weight decay strength is crucial for effective fine-tuning (green curve) of feedback-weight matching.

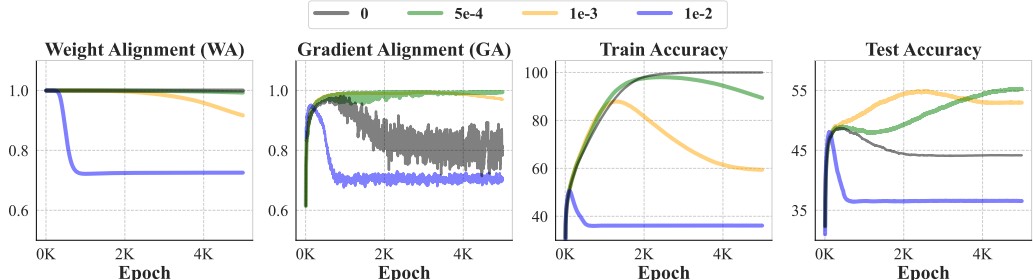

Figure 3: **WA/GA and train/test accuracy** on various weight decays (0, 5e-4, 1e-3, 1e-2). A 4-layer fully connected network is fine-tuned from CIFAR-100 to CIFAR-10 by feedback-weight matching.

## ACKNOWLEDGMENT

This work was supported by the Institute of Information & communications Technology Planning & Evaluation (IITP) grant funded by the Korea government (MSIT) (No.RS-2024-00508465 and RS-2025-25442469).

## THE USE OF LARGE LANGUAGE MODELS (LLMS)

We used large language models solely for polishing grammar and improving the readability of the manuscript. The contents and research contributions were written entirely by the authors.

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

# A  PROOF

## A.1  PROOF OF PROPOSITION 3.3

*Proof.* We prove Prop. 3.3 for $\mathbf{W}^t_{1<l<L}$ in Eq. (15), and the same reasoning applies to $\mathbf{W}^t_L$ in (16). Since $\mathbf{A}^t_{l\geq2}$ in Eq. (3) becomes such that $\mathbf{A}^t_{l\geq2} \propto \mathbf{I}$ as the training proceeds (Refinetti et al., 2021), the weight newly updated with DFA, which is denoted as $\bar{\mathbf{W}}^t_{1<l<L}$, comes to satisfy Eq. (4), i.e., $\bar{\mathbf{W}}^t_{1<l<L} = c^t_l \mathbf{F}_l \mathbf{F}^\top_{l-1}$ with some constant $c^t_l$. Given that we take the pre-trained weight $\mathbf{W}^0_{1<l<L}$ as the initial training point in our fine-tuning, the overall weight $\mathbf{W}^t_{1<l<L}$ obtained by DFA is expressed as the sum of $\mathbf{W}^0_{1<l<L}$ and $\bar{\mathbf{W}}^t_{1<l<L}$, which is given by:

$$\mathbf{W}^t_{1<l<L} = \mathbf{W}^0_{1<l<L} + \bar{\mathbf{W}}^t_{1<l<L} = \mathbf{W}^0_{1<l<L} + c^t_l \mathbf{F}_l \mathbf{F}^\top_{l-1} \not\propto \mathbf{F}_l \mathbf{F}^\top_{l-1} \tag{15}$$

$$\mathbf{W}^t_L = \mathbf{W}^0_L + \bar{\mathbf{W}}^t_L = \mathbf{W}^0_L + c^t_L \mathbf{F}^\top_{L-1} \not\propto \mathbf{F}^\top_{L-1} \tag{16}$$

where $c^t_{1<l\leq L}$ is a constant. In Eq. (15), since $\mathbf{W}^0_{1<l<L}$ is unlikely to be proportional to $\mathbf{F}_l \mathbf{F}^\top_{l-1}$, i.e., $\mathbf{W}^0_{1<l<L} \not\propto \mathbf{F}_l \mathbf{F}^\top_{l-1}$, the overall weight $\mathbf{W}^t_{1<l<L}$, which includes $\mathbf{W}^0_{1<l<L}$, is also unlikely to be proportional to $\mathbf{F}_l \mathbf{F}^\top_{l-1}$, i.e., $\mathbf{W}^t_{1<l<L} \not\propto \mathbf{F}_l \mathbf{F}^\top_{l-1}$, though $\bar{\mathbf{W}}^t_{1<l<L} = c^t_l \mathbf{F}_l \mathbf{F}^\top_{l-1} \propto \mathbf{F}_l \mathbf{F}^\top_{l-1}$. Hence, Eq. (15) can hardly induce strong WA in Eq. (4). $\square$

## A.2  PROOF OF PROPOSITION 3.6

*Proof.* Similar to Eq. (15) and (16), the overall weight $\mathbf{W}^t_l$ obtained by DFA is the sum of $\mathbf{W}^0_l$ and $\bar{\mathbf{W}}^t_l$. Specifically, now that $\bar{\mathbf{W}}^0_{1<l<L} = \bar{\mathbf{F}}_l \bar{\mathbf{F}}^\top_{l-1}$ and $\bar{\mathbf{W}}^0_L = \bar{\mathbf{F}}^\top_{L-1}$, these become proportional to $\bar{\mathbf{F}}_l \bar{\mathbf{F}}^\top_{l-1}$ and $\bar{\mathbf{F}}_{L-1}$, respectively, as:

$$\mathbf{W}^t_{1<l<L} = \bar{\mathbf{W}}^0_{1<l<L} + \bar{\mathbf{W}}^t_{1<l<L} = \bar{\mathbf{F}}_l \bar{\mathbf{F}}^\top_{l-1} + c^t_l \bar{\mathbf{F}}_l \bar{\mathbf{F}}^\top_{l-1} = (1 + c^t_l)\bar{\mathbf{F}}_l \bar{\mathbf{F}}^\top_{l-1} \propto \bar{\mathbf{F}}_l \bar{\mathbf{F}}^\top_{l-1} \tag{17}$$

$$\mathbf{W}^t_L = \bar{\mathbf{W}}^0_{\mathbf{L}} + \bar{\mathbf{W}}^t_L = \bar{\mathbf{F}}^\top_{L-1} + c^t_L \bar{\mathbf{F}}^\top_{L-1} = (1 + c^t_L)\bar{\mathbf{F}}^\top_{L-1} \propto \bar{\mathbf{F}}^\top_{L-1} \tag{18}$$

with constants $c^t_{1<l\leq L}$, which aligns with the strong WA condition in Eq. (4). $\square$

## A.3  PROOF OF PROPOSITION 3.8

*Proof.* The weight at the second layer of the network, $\mathbf{W}^t_2$, can be expressed with the pre-trained weight, $\mathbf{W}^0_2$, with the learning rate $\eta$, the number of neurons as $p$, $\mathbf{F}_1 \in R^p$, and $\mathbf{W}^t_2 \in R^p$ as follows (Song et al., 2021).

$$\mathbf{W}^t_2 = \mathbf{W}^{t-1}_2 - \eta \frac{1}{\sqrt{p}} \mathbf{W}^{t-1}_1 \mathbf{X}^\top \mathbf{e}^{t-1} = \mathbf{W}^0_2 - \frac{\eta}{\sqrt{p}} \sum_{t=0}^{t'-1} \mathbf{W}^t_1 \mathbf{X}^\top \mathbf{e}^t \tag{19}$$

For the standard DFA that does not apply feedback-weight matching in Eq. (6) and (7), we have $G_{DFA} = \mathbf{F}_1$ and $G_{BP} = \mathbf{W}^t_2$. By using Eq. (19), the gradient alignment (GA) defined in Eq. (9) between them, which is denoted as $\cos_{DFA} \angle(\mathbf{F}_1, W^t_2)$, is at least as follows.

$$\cos_{DFA} \angle(\mathbf{F}_1, \mathbf{W}^t_2) = \frac{F^\top_1 \mathbf{W}^t_2}{\|\mathbf{F}_1\|\|\mathbf{W}^t_2\|} = \frac{\frac{\mathbf{F}^\top_1}{\|\mathbf{F}_1\|} \mathbf{W}^t_2}{\|\mathbf{W}^t_2\|} = \frac{\frac{\mathbf{F}^\top_1}{\|\mathbf{F}_1\|}(\mathbf{W}^0_2 - \frac{\eta}{\sqrt{p}} \sum_{t=0}^{t'-1} \mathbf{W}^t_1 \mathbf{X}^\top \mathbf{e}^t)}{\|\mathbf{W}^0_2 - \frac{\eta}{\sqrt{p}} \sum_{t=0}^{t'-1} \mathbf{W}^t_1 \mathbf{X}^\top \mathbf{e}^t\|}$$
$$\geq \frac{\frac{\mathbf{F}^\top_1}{\|\mathbf{F}_1\|}(\mathbf{W}^0_2 - \frac{\eta}{\sqrt{p}} \sum_{t=0}^{t'-1} \mathbf{W}^t_1 \mathbf{X}^\top \mathbf{e}^t)}{\|\mathbf{W}^0_2\| + \|\frac{\eta}{\sqrt{p}} \sum_{t=0}^{t'-1} \mathbf{W}^t_1 \mathbf{X}^\top \mathbf{e}^t\|} \tag{20}$$

Conversely, when applying feedback-weight matching in Eq. (6) and (7), we have $\mathbf{F}_1 = \mathbf{W}^0_2$ for $L=2$. Using Eq. (19) again, GA between them, $\cos_{FWM} \angle(\mathbf{F}_1, \mathbf{W}^t_2)$, is at least as follows.

$$\cos_{FWM} \angle(\mathbf{F}_1, \mathbf{W}^t_2) = \frac{\frac{\mathbf{F}^\top_1}{\|\mathbf{F}_1\|}(\mathbf{W}^0_2 - \frac{\eta}{\sqrt{p}} \sum_{t=0}^{t'-1} \mathbf{W}^t_1 \mathbf{X}^\top \mathbf{e}^t)}{\|\mathbf{W}^0_2 - \frac{\eta}{\sqrt{p}} \sum_{t=0}^{t'-1} \mathbf{W}^t_1 \mathbf{X}^\top \mathbf{e}^t\|} \geq \frac{\frac{\mathbf{F}^\top_1}{\|\mathbf{F}_1\|}(\mathbf{F}_1 - \frac{\eta}{\sqrt{p}} \sum_{t=0}^{t'-1} \mathbf{W}^t_1 \mathbf{X}^\top \mathbf{e}^t)}{\|\mathbf{F}_1\| + \|\frac{\eta}{\sqrt{p}} \sum_{t=0}^{t'-1} \mathbf{W}^t_1 \mathbf{X}^\top \mathbf{e}^t\|}. \tag{21}$$

If we assume that both $\mathbf{F}_1$ and $\mathbf{W}^0_2$ follow the standard Gaussian distribution, we have $\|\mathbf{F}^\top_1 \mathbf{W}^0_2\| \leq \|\mathbf{F}_1\|^2$ (Song et al., 2021). Thus, $\cos_{FWM} \angle(\mathbf{F}_1, \mathbf{W}^t_2)$ exhibits a higher lower bound compared to $\cos_{DFA} \angle(\mathbf{F}_1, \mathbf{W}^t_2)$, i.e., $\cos_{FWM} \angle(\mathbf{F}_1, \mathbf{W}^t_2) \geq \cos_{DFA} \angle(\mathbf{F}_1, \mathbf{W}^t_2)$, implying a higher GA. $\square$

## A.4 Proof of Lemma 4.1

*Proof.* We show that $r_{1<l<L} \geq 0$ in Eq. (11), and the same reasoning extends to $r_L$ in (12). Given that $\bar{\mathbf{W}}_l^0 = \bar{\mathbf{F}}_l \bar{\mathbf{F}}_{l-1}^\top \propto \mathbf{W}_l^t = c_l^t \bar{\mathbf{F}}_l \bar{\mathbf{F}}_{l-1}^\top$, we can interpret $\mathbf{W}_l^t$ as a scaled version of $\bar{\mathbf{W}}_l^0$, which implies that $\|\mathbf{W}_l^t - \bar{\mathbf{W}}_l^0\|$ is small. Conversely, since $\mathbf{W}_l^0$ is not proportional to $\mathbf{W}_l^t$, i.e., $\mathbf{W}_l^0 \not\propto \mathbf{W}_l^t = c_l^t \bar{\mathbf{F}}_l \bar{\mathbf{F}}_{l-1}^\top$, it follows that $\|\mathbf{W}_l^t - \mathbf{W}_l^0\|$ is generally larger than $\|\mathbf{W}_l^t - \bar{\mathbf{W}}_l^0\|$. Therefore, $\|\mathbf{W}_l^t - \bar{\mathbf{W}}_l^0\|$ is likely smaller than $\|\mathbf{W}_l^t - \mathbf{W}_l^0\|$. $\square$

## A.5 Proof of Proposition 4.2

*Proof.* It is shown (Song et al., 2021) that the inequality in Eq. (13), i.e., $\|\mathbf{e}^{t+1}\| \leq \left(1 - \frac{\eta\gamma}{4} - \eta\lambda^t\right)\|\mathbf{e}^t\| + \lambda^t\|\mathbf{y}\|$, holds for a two-layer fully connected non-linear network when applying FA (Feedback Alignment) (Lillicrap et al., 2016) with weight decay (Krogh & Hertz, 1991). Specifically, the right-hand side of the inequality, i.e., $\left(1 - \frac{\eta\gamma}{4} - \eta\lambda^t\right)\|\mathbf{e}^t\| + \lambda^t\|\mathbf{y}\|$, consists of the following term as a linear component in fine-tuning:

$$\|\mathbf{W}_2^t - \mathbf{W}_2^0\| \text{ s.t. } \mathbf{W}_2^0 \not\propto \mathbf{F}_1^\top \tag{22}$$

where $\mathbf{W}_2^0$ is the pre-trained weights. By assuming that $\mathbf{W}_2^0$ is replaced with the re-initialized weights, $\bar{\mathbf{W}}_2^0$ in Eq. (7), $\|\mathbf{e}^{t+1}\|$ in Eq. (13) is reduced by $\alpha_2 r_2$ since $\|\mathbf{W}_2^t - \mathbf{W}_2^0\| \geq \|\mathbf{W}_2^t - \bar{\mathbf{W}}_2^0\|$, as in Lem. 4.1. $\square$

## B Decomposition of weight into feedback matrices

One way of finding feedback matrices $\bar{\mathbf{F}}_l$ and $\bar{\mathbf{F}}_{l-1}^\top$ in Eq. (6) from $\mathbf{W}_{1<l<L}^0$, other than SVD (Singular Value Decomposition) (Klema & Laub, 1980), is to optimize the following objective $\mathcal{L}_{FM}$.

$$\mathcal{L}_{FM} = \frac{1}{2}\sum_{l=2}^{L-1}(\mathbf{W}_l^0 \mathbf{h}_{l-1} - \bar{\mathbf{F}}_l \bar{\mathbf{F}}_{l-1}^\top \mathbf{h}_{l-1})^2 + \frac{1}{2}(\mathbf{W}_L^0 \mathbf{h}_{L-1} - \bar{\mathbf{F}}_{L-1}\mathbf{h}_{L-1})^2 + \frac{1}{2}\sum_{l=1}^{L-1}(\mathbf{I} - \bar{\mathbf{F}}_l^\top \bar{\mathbf{F}}_l)^2 \tag{23}$$

Here, $\mathcal{L}_{FM}$ is minimized to ensure that the layer output, when replaced by the feedback matrix $\bar{\mathbf{F}}_l \bar{\mathbf{F}}_{l-1}^\top \mathbf{h}_{l-1}$, matches the output obtained using the pre-trained weight $\mathbf{W}_l^0 \mathbf{h}_{l-1}$, while $\bar{\mathbf{F}}_l$ is to be orthogonal to itself in accordance with the regular DFA condition (Lillicrap et al., 2016).

## C Limitations and future works

**Extending to different architectures**. Although this study presents the significant potential of fine-tuning with DFA, its current application is restricted to fully connected networks. This limitation arises because, at present, DFA is predominantly effective for fully connected architectures, and further research is needed to extend its applicability to other network types. In our future work, we plan to explore the application of DFA fine-tuning to various network architectures, such as CNNs. Meanwhile, we anticipate the development of more generalized methods that will enable DFA to be applied across a broader range of network types, thereby enhancing the applicability of our work.

**Improving learning performance**. The learning performance of the proposed feedback-weight matching is shown to surpass both 1) training networks with DFA from scratch and 2) fine-tuning networks with DFA using random feedback matrices. While fine-tuning with DFA applying the proposed method achieves superior and more stable performance compared to them, it still falls short of the performance achieved with fine-tuning using back-propagation (Rumelhart et al., 1986). We plan to explore how to achieve fine-tuning performance comparable to that of back-propagation by investigating DFA from its fundamental mechanism, along with the proposed method.

**Proving hypotheses**. This work provides some hypotheses regarding fine-tuning and weight decay in the context of DFA. Conj. 4.3 posits that applying the proposed method to weight decay enhances fine-tuning performance of DFA for fully connected networks of arbitrary layers. However, formal proofs are necessary to substantiate these hypotheses and validate the efficacy of the proposed approach. In future research, we intend to generalize the propositions presented in this study to encompass various types of fully connected network architectures.

# D  ABLATION EXPERIMENT ON BERT

Tab. 6 presents the fine-tuning performance of BERT models when weight matching, feedback matching, and weight decay are individually removed. It is important to note that DFA is not applied to all fully connected layers in BERT, which limits the ability to properly assess the effectiveness of feedback-weight matching. Thus, this experimental setup may not provide an accurate evaluation.

Table 6: **Ablation experiment.** The fine-tuning performance when removing weight matching (DFA$_{weight*}$), feedback matching (DFA$_{feed*}$), and weight decay (DFA$_{decay*}$). 'DFA$_{ours}$' denotes applying all of them.

| Model | Training | CoLA (mat-cor) | SST-2 (acc) | MRPC (acc) | QQP (acc) | MNLI (acc) | QNLI (acc) | STSB (pearson) | RTE (acc) | WNLI (acc) |
|---|---|---|---|---|---|---|---|---|---|---|
| | DFA$_{weight*}$ | 0.00 | 94.7 | 67.4 | 81.4 | 59.2 | 88.4 | -0.15 | 50.3 | 50.9 |
| BERT-Tiny | DFA$_{feed*}$ | 0.00 | 95.8 | 68.9 | 82.4 | 60.8 | 86.9 | 0.35 | 55.5 | 50.0 |
| | DFA$_{decay*}$ | 0.31 | 95.9 | 71.4 | 81.9 | 61.0 | 83.3 | 0.36 | 53.3 | 51.9 |
| | DFA$_{ours}$ | 0.29 | 95.9 | 69.7 | 82.3 | 60.2 | 84.3 | 0.36 | 50.8 | 52.6 |
| | DFA$_{weight*}$ | 0.08 | 96.0 | 75.1 | 85.0 | 66,7 | 79.7 | -0.06 | 61.8 | 50.1 |
| BERT-Small | DFA$_{feed*}$ | 0.54 | 97.0 | 91.5 | 87.4 | 65.2 | 85.3 | 0.75 | 62.0 | 50.2 |
| | DFA$_{decay*}$ | 0.53 | 97.2 | 91.2 | 87.1 | 64.7 | 85.4 | 0.78 | 68.7 | 50.9 |
| | DFA$_{ours}$ | 0.53 | 97.3 | 92.5 | 86.9 | 65.8 | 87.2 | 0.76 | 59.0 | 51.0 |

# E  LAYER-WISE GRADIENT ALIGNMENT ANALYSIS

Weight alignment and gradient alignment have generally been analyzed in the context of sequential fully connected layers. However, Transformer-based models introduce attention mechanisms, which disrupt the sequential structure of fully connected layers. To investigate the impact of attention on our method, we analyze layer-wise gradient alignment across different components of the Transformer. Figure 4 illustrates the average gradient alignment of each sequential fully connected layer in a BERT model trained on the GLUE dataset. The key, query, and value layers in the attention module function as usual. As a result, the fully connected layer following the attention operation exhibits notably lower gradient alignment compared to others. This indicates that the attention operation interferes with gradient alignment and highlights the need for additional architectural considerations tailored to the attention module. Although our method alone improves both gradient alignment and performance in the sequential fully connected layers of Transformer models, further enhancing alignment within the attention module leads to even greater performance gains.

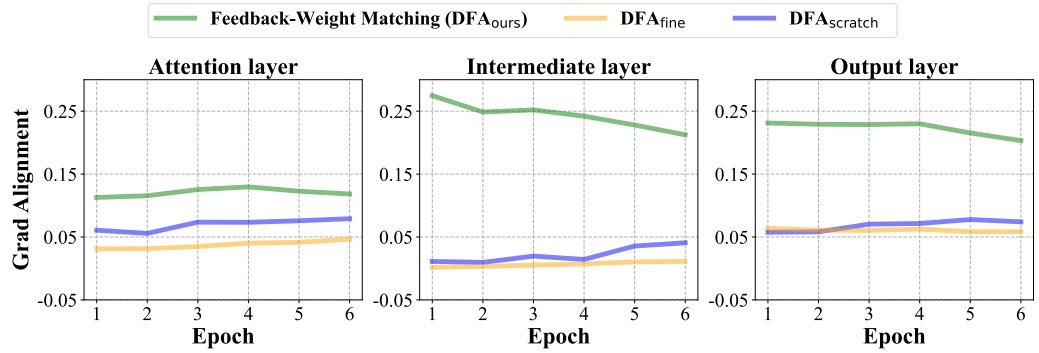

Figure 4: **Gradient alignment comparison across attention, intermediate, and output layer**. The green graph denotes DFA fine-tuning with feedback-weight matching (ours), yellow denotes DFA fine-tuning without feedback-weight matching, blue is DFA trained from scratch.

# F  COMPARISON WITH BACK-PROPAGATION

To demonstrate the effectiveness of feedback-weight matching in enhancing DFA-based fine-tuning, we presented empirical results in the previous sections. These results indicate that we have success-

fully achieved our objective of enabling and improving DFA-based fine-tuning. But, it still falls short of the performance achieved by backpropagation. In this section, we provide a detailed comparison between DFA and back-propagation in terms of fine-tuning performance. This analysis helps position DFA fine-tuning within the broader landscape of learning algorithms and clarifies its current limitations and future potential.

For a fair comparison, we use the same pre-trained weights and experimental setup as in the DFA experiments. The experimental details are in Sec. I. We denote fine-tuning with backpropagation as $BP_{fine}$, and training from scratch with backpropagation as $BP_{scratch}$. The performance of feedback-weight matching does not yet reach that of back-propagation fine-tuning. However, it is often comparable to, and occasionally surpasses, the performance of back-propagation trained from scratch. For instance, in the case of training on STL-10 with a fully connected architecture, feedback-weight matching can outperform back-propagation from scratch.

Table 7: **Image classification tasks with back-propagation.** The fine-tuning performance of feedback-weight matching ($DFA_{ours}$) on the 4 and 6-layer fully connected networks, compared with standard DFA fine-tuning ($DFA_{fine}$) and back-propagation fine-tuning ($BP_{fine}$). The pre-trained weights are obtained through back-propagation (BP). For reference, we also present the from-scratch-training results of back-propagation ($BP_{scratch}$) and DFA ($DFA_{scratch}$). The bold indicates the best performance in DFA fine-tuning.

| Model | Target Data | Source Data | | | | | | | |
|---|---|---|---|---|---|---|---|---|---|
| | | Scratch | | CIFAR-100 | | | TinyImageNet | | |
| | | $BP_{scratch}$ | $DFA_{scratch}$ | $BP_{fine}$ | $DFA_{fine}$ | $DFA_{ours}$ | $BP_{fine}$ | $DFA_{fine}$ | $DFA_{ours}$ |
| | CIFAR-10 | 55.48 | 52.78 | 57.16 | 53.79 | **55.38** | 57.66 | **56.75** | 55.51 |
| 4 layers | SVHN | 85.10 | 82.93 | 84.32 | 79.55 | **82.87** | 84.69 | 80.31 | **83.16** |
| | STL-10 | 43.15 | 42.20 | 47.73 | 44.83 | **45.30** | 50.29 | **50.62** | 45.61 |
| | CIFAR-10 | 54.93 | 51.94 | 58.85 | 53.04 | **55.39** | 55.97 | 51.08 | **55.54** |
| 6 layers | SVHN | 85.10 | 81.89 | 84.34 | 74.70 | **82.67** | 84.72 | 76.03 | **81.39** |
| | STL-10 | 43.10 | 40.48 | 47.78 | 43.42 | **45.28** | 47.63 | 43.33 | **45.21** |

Table 8: **NLP tasks with back-propagation.** The fine-tuning performance of feedback-weight matching ($DFA_{ours}$) on Transformer architectures (i.e., BERT-Tiny and BERT-Small), compared with standard DFA fine-tuning ($DFA_{fine}$) and back-propagation-based fine-tuning ($BP_{fine}$). The pre-trained weights are obtained via back-propagation (BP). For reference, we also present the from-scratch-training results of back-propagation ($BP_{scratch}$) and DFA ($DFA_{scratch}$). The bold indicates the best performance in DFA fine-tuning.

| Model | Training | CoLA (mat-cor) | SST-2 (acc) | MRPC (acc) | QQP (acc) | MNLI (acc) | QNLI (acc) | STSB (pearson) | RTE (acc) | WNLI (acc) |
|---|---|---|---|---|---|---|---|---|---|---|
| | $BP_{scratch}$ | 0.07 | 96.3 | 67.4 | 82.8 | 63.4 | 89.2 | -0.19 | 64.1 | 50.0 |
| | $BP_{fine}$ | 0.00 | 93.5 | 70.7 | 86.9 | 73.8 | 88.2 | -0.25 | 60.3 | 52.6 |
| BERT-Tiny | $DFA_{scratch}$ | 0.00 | 95.2 | 67.4 | 81.2 | 59.2 | 84.2 | -0.11 | 50.2 | 50.0 |
| | $DFA_{fine}$ | 0.00 | 92.4 | 67.4 | 80.6 | 60.0 | 80.2 | -0.17 | 51.2 | 51.0 |
| | $DFA_{ours}$ | **0.29** | **95.9** | **69.7** | **82.3** | **60.2** | **84.3** | **0.36** | **55.5** | **52.6** |
| | $BP_{scratch}$ | 0.55 | 96.3 | 95.4 | 91.3 | 75.3 | 93.4 | 0.67 | 89.8 | 51.9 |
| | $BP_{fine}$ | 0.87 | 98.9 | 96.7 | 98.0 | 93.0 | 99.1 | 0.90 | 94.0 | 53.3 |
| BERT-Small | $DFA_{scratch}$ | 0.19 | 96.5 | 75.2 | 86.7 | 67.4 | 80.9 | 0.05 | 60.0 | 50.3 |
| | $DFA_{fine}$ | 0.06 | 95.6 | 70.9 | 86.0 | **67.0** | 85.3 | 0.10 | 59.0 | 49.3 |
| | $DFA_{ours}$ | **0.53** | **97.3** | **92.5** | **86.9** | 65.8 | **87.2** | **0.76** | **59.0** | **51.0** |

## G  RECOVERY ABILITY OF THE FEEDBACK MATRICES

We evaluate the resilience of feedback matrices in the Feedback-Weight Matching process. The feedback matrices are designed to mimic the existing pre-trained weights and preserve their information. Accordingly, we examine the extent of performance degradation when re-initialized weights, generated using the trained feedback matrices, are evaluated on the pre-trained dataset CIFAR-100. Tab. 9 reports the performance of pre-trained weights and re-initialized weights obtained from the trained feedback matrices on CIFAR-100. For most classification tasks, the layer dimensions are larger than the number of classes. Consequently, feedback matrices with dimensions equal to the

Table 9: Comparison between Pre-trained and FWM weights on CIFAR-100

|  | Pre-trained Weight | FWM Weight |
|---|---|---|
| CIFAR-100 | 32.38 | 26.86 |

Table 10: Results according to epoch within the Feedback-Matching process

| Dataset | $DFA_{low}$ | $DFA_{mid}$ | $DFA_{ours}$ |
|---|---|---|---|
| CIFAR-10 | 51.38 | 53.27 | 55.38 |

number of classes have a lower rank than the pre-trained weights. This inevitably leads to some loss of information, resulting in slightly reduced performance of the re-initialized weights compared to the pre-trained weights.

To investigate how the training of feedback matrices affects fine-tuning performance, we measured fine-tuning results at different epochs in the Feedback-Matching process. In Tab. 10, $DFA_{low}$ corresponds to 1 epoch, $DFA_{mid}$ to 2 epochs, and $DFA_{ours}$ to 3 epochs of feedback matrix training, which matches the original setting. When the feedback matrices are less thoroughly trained, fine-tuning performance decreases. This indicates that insufficiently trained feedback matrices fail to capture the information of the pre-trained weights, leading to reduced performance.

## H  TRAINING COST

The training cost of our method does not differ significantly from that of standard DFA, as training proceeds using standard DFA after re-initializing the feedback matrices and weights. Standard DFA theoretically enables layerwise parallel training and can achieve a speed-up proportional to the number of layers compared to back-propagation, but improving its training efficiency is not the focus of this work. Consequently, during the fine-tuning phase, training speed and memory overhead remain equivalent to those of standard DFA. Furthermore, the overhead introduced by the feedback matching stage is minimal, as it is conducted for only three epochs.

Although DFA theoretically supports layer-parallel updates, implementing true parallelism within a single model remains challenging. Existing parallelization methods mainly focus on distributing data or model components across multiple GPUs, whereas they do not readily support inter-layer parallel execution on a single GPU. In our implementation, we approximate the behavior of DFA by splitting the backward graph at each layer and substituting the local gradient with the DFA error signal. This design preserves the learning dynamics of DFA, yet the update process still proceeds sequentially as in standard back-propagation.

To illustrate the potential efficiency gains of parallel DFA, we report per-layer backward computation times measured on a six-layer MLP with 1000 hidden units trained on CIFAR-100 in Tab. 11. In back-propagation, lower layers must wait for the computations of upper layers, which results in cumulative backward times. In contrast, DFA can, in principle, update all layers simultaneously so that the total update time would be determined only by the slowest layer. Although full parallelization is not implemented, the per-layer measurements provide an upper bound on the possible speed-up.

Table 11: **DFA vs BP layer-wise and total backward time comparison (ms)**. Layer-wise values indicate the backward computation time for each layer, whereas total values represent accumulated backward time for BP and the maximum layer time for DFA under ideal parallel execution. Results are measured on a six-layer MLP with 1,000 hidden units trained on CIFAR-100. The maximum DFA layer time is highlighted.

| Type | layers 1 | layers 2 | layers 3 | layers 4 | layers 5 | layers 6 |
|---|---|---|---|---|---|---|
| DFA (layer) | 0.0132 | **0.1003** | 0.0942 | 0.0910 | 0.0895 | 0.0811 |
| BP (layer) | 0.0098 | 0.0899 | 0.0927 | 0.0942 | 0.0957 | 0.0996 |
| DFA (total) | **0.1003** | **0.1003** | **0.1003** | **0.1003** | **0.1003** | **0.1003** |
| BP (total) | 0.4820 | 0.4722 | 0.3823 | 0.2895 | 0.1953 | 0.0996 |

Regarding memory usage, DFA requires additional feedback matrices of size $N_l \times e$ for each layer, where $N_l$ is the number of neurons and $e$ is the dimensionality of the error vector. As a result, DFA exhibits higher memory consumption compared to back-propagation. In our experiments, DFA reached a peak memory usage of approximately 130.99 MiB, while back-propagation used around 100.75 MiB. This difference reflects the cost of storing the feedback matrices. From a computational perspective, DFA also differs from back-propagation in terms of the operations required for propagating error signals. Back-propagation computes $W_{l+1}\delta a_{l+1}$, which requires $\mathcal{O}(N_l N_{l+1})$ operations, whereas DFA computes $F_l e$, requiring $\mathcal{O}(N_l N_L)$. Since $N_L$ is typically much smaller than $N_{l+1}$ in classification tasks, DFA can require fewer operations and reduced data movement. Although exploiting this theoretical efficiency is not the main objective of our study, we include this analysis for completeness.

## I    EXPERIMENTAL SETUPS

In this section, we offer an explanation of the experimental setup utilized throughout our research. Sec. I.1 outlines the training details of the feedback matrix used for feedback matching in all models. Sec. I.2 covers the configuration settings required for the fully connected network experiments. Sec. I.3 describes the setup necessary for experiments involving BERT, which employs a transformer architecture. Sec. I.4 provides the setup employed for experiments with the ViT model. For fine-tuning of BERT and ViT, feedback-weight matching is applied to the attention, intermediate, and block outputs of the encoder layers in a similar way to previous works (Launay et al., 2020) that attempt to apply DFA to Transformer's attention modules (Vaswani, 2017). To ensure the robustness of our findings, we report the average results over three different random seeds. All experiments were conducted on an NVIDIA GeForce RTX 3090 GPU with 24GB of memory.

### I.1    FEEDBACK MATRIX

We train feedback matrices to reconstruct pre-trained weights that were trained using back-propagation (Rumelhart et al., 1986). The loss function, in Eq. (23), is used to guide the feedback matching process. The two learned feedbacks are then combined and re-initialized into a single weight matrix for each layer. We use the Adam optimizer (Kingma, 2014) without weight decay or any scheduler. In fully connected networks, a learning rate of 1e-5 is applied, while in transformers (BERT) (Kenton & Toutanova, 2019; Turc et al., 2019), a learning rate of 1e-3 is used. For all experiments on the model and dataset, training is conducted for 3 epochs with a batch size of 64.

### I.2    FULLY CONNECTED NETWORKS

We pre-train two fully connected networks with four and six layers on the CIFAR-100 (Krizhevsky et al., 2009) and TinyImageNet (Le & Yang, 2015) datasets utilizing weights obtained through back-propagation (BP). These pre-trained weights are subsequently fine-tuned on the CIFAR-10 (Krizhevsky et al., 2009), SVHN (Netzer et al., 2011), and STL-10 (Coates et al., 2011) datasets. During the pre-processing phase, we apply image resizing and normalization, without any augmentations. For Direct Feedback Alignment (DFA) (Nøkland, 2016), the weights are initialized with a uniform distribution within the range of (-0.01, 0.01). Conversely, for back-propagation (Rumelhart et al., 1986), we employ the He initialization (He et al., 2015). The optimization process is carried out using Stochastic Gradient Descent, and ReLU (Agarap, 2018) is employed as the activation function. The hyperparameters for both the 4-layer and 6-layer architectures remain consistent. A comprehensive description of each hyperparameter under various training conditions is presented in Tab. 12.

### I.3    BERT

We train BERT-Tiny and Small models (Kenton & Toutanova, 2019; Turc et al., 2019) on the GLUE (Wang, 2018) dataset using the AdamW (Loshchilov, 2017) optimizer with a fixed learning rate and no scheduler. We apply weight decay and dropout techniques. GeLU (Hendrycks & Gimpel, 2016) is used for the activation function, which is commonly employed in BERT. Layers such as the encoder block outputs, intermediate outputs, and attention outputs are optimized using Direct Feedback Alignment (DFA) (Nøkland, 2016), while the projection layers for key, query, and

Table 12: **Hyperparameters for fully connected networks training.**

| Target Data | Hyperparmeters | BP$_{scratch}$ | BP$_{fine}$ | DFA$_{scratch}$ | DFA$_{fine}$ | DFA$_{feed}$ | DFA$_{weight}$ | DFA$_{ours}$ |
|---|---|---|---|---|---|---|---|---|
| | Learning Rate | 1e-3 | 1e-3 | 1e-3 | 1e-3 | 1e-3 | 1e-3 | 1e-3 |
| | Batch size | 64 | 64 | 64 | 64 | 64 | 64 | 64 |
| | Hidden Dim | 1000 | 1000 | 1000 | 1000 | 1000 | 1000 | 1000 |
| | Input size | 3072 | 3072 | 3072 | 3072 | 3072 | 3072 | 3072 |
| | Epochs | 5000 | 5000 | 5000 | 5000 | 5000 | 5000 | 5000 |
| CIFAR-10 | Weight Decay | 5e-4 | 5e-4 | 0 | 0 | 5e-4 | 5e-4 | 5e-4 |
| | Dropout | 0.1 | 0.1 | 0 | 0 | 0 | 0 | 0 |
| | Epochs | 5000 | 5000 | 5000 | 5000 | 5000 | 5000 | 5000 |
| SVHN | Weight Decay | 5e-4 | 5e-4 | 0 | 0 | 5e-4 | 5e-4 | 5e-4 |
| | Dropout | 0.1 | 0.1 | 0 | 0 | 0 | 0 | 0 |
| | Epochs | 5000 | 5000 | 5000 | 5000 | 30000 | 30000 | 30000 |
| STL-10 | Weight Decay | 5e-4 | 5e-4 | 0 | 0 | 1e-3 | 1e-3 | 1e-3 |
| | Dropout | 0.1 | 0.1 | 0 | 0 | 0.1 | 0.1 | 0.1 |

value are trained using back-propagation (BP) (Rumelhart et al., 1986). The weights are initialized using a uniform distribution, and the feedback matrix is specifically designed to satisfy the left orthogonality condition. A comprehensive description of the hyperparameter values is presented in Tab. 13.

Table 13: **Hyperparameters for BERT training.**

| Model | Hyperparmeters | Target Data | BP$_{scratch}$ | BP$_{fine}$ | DFA$_{scratch}$ | DFA$_{fine}$ | DFA$_{feed}$ | DFA$_{weight}$ | DFA$_{ours}$ |
|---|---|---|---|---|---|---|---|---|---|
| | Batch size | | 64 | 64 | 64 | 64 | 64 | 64 | 64 |
| | Dropout | | 0.1 | 0.1 | 0.1 | 0.1 | 0.1 | 0.1 | 0.1 |
| | Weight Decay | | 0.01 | 0.01 | 0.01 | 0.01 | 0.01 | 0.01 | 0.01 |
| | Epochs | | 6 | 6 | 6 | 6 | 6 | 6 | 6 |
| | Max length | | 512 | 512 | 512 | 512 | 512 | 512 | 512 |
| | Num of heads | | 2 | 2 | 2 | 2 | 2 | 2 | 2 |
| | Num of layers | | 2 | 2 | 2 | 2 | 2 | 2 | 2 |
| | Hidden dim | | 128 | 128 | 128 | 128 | 128 | 128 | 128 |
| | Intermediate dim | | 512 | 512 | 512 | 512 | 512 | 512 | 512 |
| | | CoLA | 1e-5 | 1e-5 | 1e-5 | 1e-5 | 1e-5 | 1e-5 | 1e-5 |
| | | SST-2 | 1e-5 | 1e-5 | 1e-5 | 1e-5 | 1e-5 | 1e-5 | 1e-5 |
| | | MRPC | 1e-5 | 1e-5 | 1e-5 | 1e-5 | 1e-5 | 1e-5 | 1e-5 |
| | | QQP | 1e-5 | 1e-5 | 1e-5 | 1e-5 | 1e-5 | 1e-5 | 1e-5 |
| BERT-Tiny | Learning Rate | MNLI | 1e-5 | 1e-5 | 1e-5 | 1e-5 | 1e-5 | 1e-5 | 1e-5 |
| | | QNLI | 5e-5 | 5e-5 | 5e-5 | 5e-5 | 5e-5 | 5e-5 | 5e-5 |
| | | STSB | 1e-5 | 1e-5 | 1e-5 | 1e-5 | 1e-5 | 1e-5 | 1e-5 |
| | | RTE | 1e-5 | 1e-5 | 1e-5 | 1e-5 | 1e-5 | 1e-5 | 1e-5 |
| | | WNLI | 5e-5 | 5e-5 | 5e-5 | 5e-5 | 5e-5 | 5e-5 | 5e-5 |
| | Num of heads | | 8 | 8 | 8 | 8 | 8 | 8 | 8 |
| | Num of layers | | 4 | 4 | 4 | 4 | 4 | 4 | 4 |
| | Hidden of dim | | 512 | 512 | 512 | 512 | 512 | 512 | 512 |
| | Intermediate dim | | 2048 | 2048 | 2048 | 2048 | 2048 | 2048 | 2048 |
| | | CoLA | 1e-5 | 1e-5 | 1e-5 | 1e-5 | 1e-5 | 1e-5 | 1e-5 |
| | | SST-2 | 1e-5 | 1e-5 | 1e-5 | 1e-5 | 1e-5 | 1e-5 | 1e-5 |
| BERT-Small | | MRPC | 1e-5 | 1e-5 | 1e-5 | 1e-5 | 1e-5 | 1e-5 | 1e-5 |
| | | QQP | 1e-5 | 1e-5 | 1e-5 | 1e-5 | 1e-5 | 1e-5 | 1e-5 |
| | Learning Rate | MNLI | 1e-5 | 1e-5 | 1e-5 | 1e-5 | 1e-5 | 1e-5 | 1e-5 |
| | | QNLI | 5e-5 | 5e-5 | 5e-5 | 5e-5 | 5e-5 | 5e-5 | 5e-5 |
| | | STSB | 1e-5 | 1e-5 | 1e-5 | 1e-5 | 1e-5 | 1e-5 | 1e-5 |
| | | RTE | 1e-5 | 1e-5 | 1e-5 | 1e-5 | 1e-5 | 1e-5 | 1e-5 |
| | | WNLI | 1e-5 | 1e-5 | 1e-5 | 1e-5 | 1e-5 | 1e-5 | 1e-5 |

## I.4 VIT

We fine-tune ViT-Tiny and Small models (Wu et al., 2022), both pre-trained on ImageNet-1K (Deng et al., 2009), on the CIFAR-10 (Krizhevsky et al., 2009), STL-10 (Coates et al., 2011), and ImageNette (Howard, 2019) datasets. For preprocessing, we resize the 32x32 images from CIFAR-10 and STL-10 to 224x224 and apply normalization. We use the AdamW (Loshchilov, 2017) optimizer and GeLU (Hendrycks & Gimpel, 2016) as the activation function. Following the approach used in

BERT, we apply Direct Feedback Alignment (DFA) to train the ViT models, specifically targeting the encoder block outputs, intermediate outputs, and attention outputs. While the Tiny and Small models have the same number of layers, they differ in the number of channels and attention heads. A comprehensive list of hyperparameters for these models is provided in Tab. 14.

Table 14: **Hyperparameters for ViT training.**

| Model | Hyperparmeters | Target Data | DFA$_{scratch}$ | DFA$_{fine}$ | DFA$_{ours}$ |
|---|---|---|---|---|---|
| | Batch size | | 64 | 64 | 64 |
| | Dropout | | 0.1 | 0.1 | 0.1 |
| | Weight Decay | | 0.01 | 0.01 | 0.01 |
| | Epochs | | 5 | 5 | 5 |
| | Image Size | | 224 | 224 | 224 |
| | Patch Size | | 16 | 16 | 16 |
| | Num of layers | | 12 | 12 | 12 |
| | | CIFAR-10 | 2e-5 | 2e-5 | 2e-5 |
| | Learning Rate | STL-10 | 2e-5 | 2e-5 | 2e-5 |
| | | ImageNette | 2e-5 | 2e-5 | 2e-5 |
| | Num of heads | | 3 | 3 | 3 |
| ViT-Tiny | Hidden dim | | 192 | 192 | 192 |
| | Intermediate dim | | 768 | 768 | 768 |
| | Num of heads | | 6 | 6 | 6 |
| ViT-Small | Hidden of dim | | 384 | 384 | 384 |
| | Intermediate dim | | 1536 | 1536 | 1536 |

