# OpenReview forum: "Enabling Fine-Tuning of Direct Feedback Alignment via Feedback-Weight Matching"
_ICLR.cc/2026/Conference — ICLR 2026 Poster_

### Official Review · Reviewer_Bsfv · 2025-10-30

**Soundness:** 3
**Presentation:** 3
**Contribution:** 2
**Rating:** 4
**Confidence:** 4

**Summary:**

This paper studies why Direct Feedback Alignment (DFA) struggles to fine-tune networks that were pre-trained with back-propagation (BP), and proposes a simple two-phase remedy called feedback-weight matching (FWM). The authors analyze DFA through Weight Alignment (WA) and Gradient Alignment (GA), arguing that randomly chosen feedback matrices prevent strong WA/GA when starting from BP weights. FWM first reconstructs feedback matrices from the BP weights ("feedback matching"), then re-initializes the weights to match those feedback matrices ("weight matching"), so that fine-tuning with standard DFA naturally induces strong WA/GA. They further prove that, together with weight decay, FWM reduces output error for two-layer settings and conjecture an extension to deeper nets. Experiments show consistent gains across FCNs, BERT (Tiny/Small), and ViT (Tiny/Small) models.

**Strengths:**

- **Clear problem framing & analysis.** The paper isolates a concrete failure mode of DFA fine-tuning from BP starts via WA/GA and proves why random feedback breaks alignment, then shows how FWM induces strong WA/GA. The link from theory (Defs./Props.) to the empirical WA/GA plots is tight and instructive for the learning-dynamics community.

- **Simple, general mechanism.** FWM is architecturally light: decompose BP weights to build feedback, then re-initialize weights to match feedback; standard DFA then applies unchanged. This “match-then-tune” recipe is easy to implement and does not require modifying optimizers or loss functions, making it appealing for practitioners exploring BP-free training.

- **Broad empirical scope.** Results span FCNs, BERT Tiny/Small, ViT Tiny/Small over CIFAR-10/SVHN/STL-10/ImageNette and GLUE tasks, consistently beating standard DFA fine-tuning and demonstrating stability improvements (WA/GA trends).

**Weaknesses:**

(1) **Limited transformer alignment coverage.** For transformers, alignment is only applied to dense layers; attention projections (Q/K/V) remain unaligned, which may cap attainable GA/WA. The paper mentions this but lacks systematic analysis of how much each component limits end-to-end gains, especially on larger ViTs or LLMs. Scaling experiments or per-block ablations would clarify.

(2) **Baselines and practical metrics.** The main comparator is standard DFA fine-tuning (and DFA-scratch). Stronger practice-oriented baselines (e.g., BP fine-tuning, BP with adapters/LoRA, partial-BP variants) and wall-clock, memory, and parallelism measurements are missing; these are crucial to justify DFA-based fine-tuning in real systems beyond accuracy alone

**Questions:**

Could the authors consider adding head-to-head comparisons against BP fine-tuning, LoRA/adapters, and partial-BP under matched budgets (same epochs/tokens, batch size, optimizer, augmentations). Report accuracy, calibration (ECE), robustness (seed/prompt sensitivity if applicable), and costs: wall-clock to target accuracy, tokens/sec or images/sec, peak memory/VRAM, and energy/FLOPs. Include at least one mid-scale vision task (e.g., ImageNet-1k fine-tuning) and NLP task (GLUE).

---

> ### Author Response · Authors · 2025-11-19
>
> We thank the reviewer for the insightful and detailed feedback. We are pleased that our analysis and proposed solutions were found meaningful, and we provide point-by-point responses to all raised concerns.
>
> ---
>
> **Weaknesses 1.**
>
> Transformer-based models such as ViT and BERT are composed of repeatedly stacked blocks with identical structure. In this work, we analyzed alignment formation by measuring WA and GA for the dense layers within each block. However, as the reviewer pointed out, we did not include a analysis of the attention projections (Q, K, V), and we acknowledge that this remains insufficient.
>
> To apply FWM to the attention projections, it is necessary to first understand the convergence behavior of the Q/K/V weights. The current FWM framework is designed under the assumption that the weights of consecutive fully connected layers converge to certain values, which enables their alignment with the feedback matrix to strengthen during training. Extending this principle to attention blocks requires analyzing how the Q, K, and V weights converge, as well as determining where the error signal should be injected (i.e., individually into each projection versus after the attention operation). These choices may influence how FWM should be applied.
>
> Therefore, additional investigation into the structural characteristics of attention blocks and their gradient flow is required, and this falls outside the scope of the current study. In future work, we plan to explore how FWM can be applied to the attention projections and, based on this, conduct a more comprehensive alignment analysis.
>
> **Weaknesses 2.**
>
> We thank the reviewer for the suggestion. Our proposed method performs the same operations as DFA during training, with the only difference being the initial values of the weights and feedback matrices; the training behavior itself remains identical to standard DFA. Therefore, our experiments primarily focused on evaluating the performance of FWM and its impact on alignment properties. The efficiency of DFA has been theoretically analyzed in prior work with respect to data movement and computation [1]. Due to the current difficulty of implementing layer-parallel updates, we compared computational efficiency at the layer level rather than measuring total training time.
>
> Table 1 shows the backward computation time for each layer in a model with six layers, each having 1,000 hidden units, trained on CIFAR-100. The total time represents the overall time required to update all layers. In BP, computations must be performed sequentially, so the times of lower layers are included, whereas in DFA, parallel computation allows these times to be independent. Regarding memory usage, the DFA model requires approximately 38.93 MiB for the model and reaches a peak memory of 130.99 MiB. In comparison, BP uses 27.02 MiB for the model and has a peak memory of 100.75 MiB. The higher memory usage in DFA is due to the additional feedback matrices.
>
> **Table 1. DFA vs BP Layer-wise and Total Backward Time Comparison (ms), with DFA max highlighted.**
>
> | Type          | layers.0 | layers.1 | layers.2 | layers.3 | layers.4 | layers.5 |
> |---------------|----------|----------|----------|----------|----------|----------|
> | DFA (layer)   | 0.0132   | **0.1003**   | 0.0942   | 0.0910   | 0.0895   | 0.0811   |
> | BP (layer)    | 0.0098   | 0.0899   | 0.0927   | 0.0942   | 0.0957   | 0.0996   |
> | DFA (total)   | **0.1003**   | **0.1003**   | **0.1003**   | **0.1003**   | **0.1003**   | **0.1003**   |
> | BP (total)    | 0.4820   | 0.4722   | 0.3823   | 0.2895   | 0.1953   | 0.0996   |
>
> Nonetheless, we note that these results alone do not fully demonstrate the advantages of DFA. Implementing the layer-parallel updates required by DFA is currently challenging. To address this, we implemented DFA by cutting the backward graph at each layer in BP and replacing the error signal. In this setting, updates proceed sequentially. Therefore, in theory, if parallel updates were implemented, the training time of a DFA model could be approximately equal to the time required for the slowest layer. If layer-parallel updates are implemented in the discussion time, we will include experiments measuring wall-clock time, memory usage, and parallelism.
>
> [1] Crafton, Brian, et al. ”Direct feedback alignment with sparse connections for local learning.”

---

> > ### Author Response · Authors · 2025-11-19
> >
> > **Weaknesses 3.**
> >
> > We thank the reviewer for the suggestion. Comparisons against BP fine-tuning can be partially observed through our experiments on ViT and BERT models. Regarding parameter-efficient methods such as LoRA or adapters, it is less clear what additional insight such comparisons would provide in the context of our study, which focuses on BP-free fine-tuning. Most existing BP-free algorithms have been evaluated only on fully connected layers, and it remains uncertain how well they extend to Transformer architectures. Nonetheless, we will attempt such comparisons within the scope of the current rebuttal.
> >
> > While measuring efficiency to clearly demonstrate the advantages of DFA is challenging under our current setup, we will provide preliminary comparisons in terms of training time, memory usage, and throughput to give a rough sense of the computational characteristics.

---

### Official Review · Reviewer_HVYL · 2025-10-31

**Soundness:** 3
**Presentation:** 3
**Contribution:** 3
**Rating:** 6
**Confidence:** 2

**Summary:**

The paper proposes feedback-weight matching (FWM) to enable reliable fine-tuning with Direct Feedback Alignment (DFA). The method reconstructs feedback matrices from pre-trained weights and then re-initializes weights to match them, yielding stronger weight/gradient alignment and improved performance; the authors also analyze how weight decay synergizes with FWM. Empirically, FWM outperforms standard DFA across FC nets, BERT-Tiny/Small, and ViT-Tiny/Small (e.g., +7.97% accuracy on SVHN and +0.66 Pearson on STS-B).

**Strengths:**

1. Simple, principled mechanism with theory: FWM is conceptually clean (feedback reconstruction + weight matching) and is supported by propositions showing it induces strong WA→GA; the paper further proves a GA advantage in a two-layer setting.

2. Consistent empirical gains across modalities/models: Improvements appear on FC image classifiers, GLUE with BERT-Tiny/Small, and ViT variants, often turning DFA fine-tuning from unreliable to usable (e.g., STS-B 0.76 vs 0.10).

3. Thorough ablations and interaction with weight decay: Ablations isolate feedback matching, weight matching, and weight decay, showing their contributions and a clear synergy between FWM and weight decay.

**Weaknesses:**

1. Scope and scale: Experiments focus on FC nets and compact Transformers/ViTs; CNNs are acknowledged as challenging for DFA, and there are no results on larger modern LMs, limiting claims of broad applicability.

2. Efficiency evidence is missing: While DFA’s appeal is “backprop-free” and parallelizable, the paper reports no wall-clock time, memory, or throughput comparisons vs BP-based fine-tuning, so the practical benefit remains speculative.

3. Theory primarily for simplified settings: Key guarantees (e.g., GA improvement) are proved for two-layer/linear cases, and extension to deep non-linear networks is left as a conjecture, weakening theoretical support for the full experimental regime.

**Questions:**

refer to weaknesses

---

> ### Author Response · Authors · 2025-11-19
>
> We appreciate the reviewer’s thoughtful evaluation and constructive suggestions. We are pleased that the reviewer finds our work timely and well-supported by evidence. In the following, we provide detailed responses to the questions and concerns raised.
>
> ---
>
> **Weaknesses 1.**
>
> In fact, previous DFA studies have mostly conducted experiments only on fully connected layers. One reason for this is that convolutional layers tend to fail at learning spatial structures under DFA, creating bottlenecks [1][2]. While there have been studies showing that DFA can also be applied to Transformer-based models[3], these works focused more on demonstrating the feasibility of training rather than achieving performance improvements.
> Based on these prior findings, we consider the ability to fine-tune Transformer models and achieve performance improvements to be a significant advancement. However, we acknowledge that our work still has limitations. In particular, experiments on a wider variety of modern CNN and Transformer models are lacking, and it would be necessary to consider expanding our experiments in this direction. In future work, we believe that by taking attention and alignment into account in Transformer models and improving the corresponding matching methods, it will be possible to achieve performance gains in deeper Transformer architectures.
>
> [1] Crafton, Brian, et al. "Direct feedback alignment with sparse connections for local learning.”
>
> [2] Launay, Julien, Iacopo Poli, and Florent Krzakala. "Principled training of neural networks with direct feedback alignment.”
>
> [3] Launay, Julien, et al. "Direct feedback alignment scales to modern deep learning tasks and architectures."
>
> **Weaknesses 2.**
>
> Our proposed method performs the same operations as standard DFA during training. The only difference is in the initialization of the weights and feedback matrices; the training behavior itself is identical to DFA. Therefore, we primarily focused on evaluating the performance of the FWM method and its effect on alignment properties.
>
> Implementing layer-parallel updates, as required by the DFA training algorithm, is currently challenging. To address this, we approximated DFA by splitting the backward graph at each layer in the BP algorithm and replacing it with the error signal. In this setup, parallel updates are not possible, and updates proceed sequentially, as in standard BP. By reporting per-layer computation times, we can indirectly illustrate the potential efficiency gains under parallel training.
>
> Table 1 shows the backward computation time for each layer in a model with six layers, each having 1,000 hidden units, trained on CIFAR-100. The total time represents the overall time required to update all layers. In BP, computations must be performed sequentially, so the times of lower layers are included, whereas in DFA, parallel computation allows these times to be independent.
>
> Regarding memory usage in this experiment, the DFA model requires approximately 38.93 MiB for the model and reaches a peak memory of 130.99 MiB. In comparison, BP uses 27.02 MiB for the model and has a peak memory of 100.75 MiB. The higher memory usage in DFA is due to the additional feedback matrices.
>
> **Table 1. DFA vs BP Layer-wise and Total Backward Time Comparison (ms), with DFA max highlighted.**
>
> | Type          | layers.0 | layers.1 | layers.2 | layers.3 | layers.4 | layers.5 |
> |---------------|----------|----------|----------|----------|----------|----------|
> | DFA (layer)   | 0.0132   | **0.1003**   | 0.0942   | 0.0910   | 0.0895   | 0.0811   |
> | BP (layer)    | 0.0098   | 0.0899   | 0.0927   | 0.0942   | 0.0957   | 0.0996   |
> | DFA (total)   | **0.1003**   | **0.1003**   | **0.1003**   | **0.1003**   | **0.1003**   | **0.1003**   |
> | BP (total)    | 0.4820   | 0.4722   | 0.3823   | 0.2895   | 0.1953   | 0.0996   |
>
> When comparing the theoretical efficiency of BP and DFA, DFA has advantages in terms of fewer operations and reduced data movement compared to backpropagation[1]. According to Equations (1) and (2) in the paper, the difference in computation between backpropagation and DFA lies in $W_{l+1}\delta a_{l+1}$ versus $F_l e$. Let $N_l$ denote the number of neurons in each layer, and we compare only this difference. Backpropagation requires $\mathcal{O}(N_l N_{l+1})$ operations, while DFA requires $\mathcal{O}(N_l N_L)$ operations. In classification tasks, the neurons in the last layer correspond to the number of classes, and $N_L$ is typically smaller than $N_l$, so DFA requires fewer operations.
>
> However, from a memory perspective, DFA requires additional feedback matrices[1]. Let $N_l$ denote the number of neurons in each layer; then these matrices must have size $N_l e$ for each layer, where $e$ is the length of the error vector. This results in higher memory usage compared to BP.

---

> > ### Author Response · Authors · 2025-11-19
> >
> > **Weaknesses 3.**
> >
> > We established theoretical results for strong weight alignment (WA), strong gradient alignment (GA), and the influence of weight decay only in two-layer neural networks. These results have not yet been formally extended to deeper architectures. However, by performing experiments on networks with more layers and nonlinear components, we aim to demonstrate that our method is applicable well beyond the narrow setting considered in our theoretical analysis.
> >
> > Figure 2 shows that incorporating weight decay alongside our method during DFA fine-tuning leads to substantial improvements in both WA and GA compared to standard DFA. In addition, Table 2 presents WA and GA measurements for a six-layer fully connected network trained on CIFAR-10.
> >
> > **Table 2. Weight Alignment and Gradient Alignment at Layer 2 in a Six-Layer Network**
> > | Layer 2 | DFA_Scratch | DFA_Fine | DFA_Ours |
> > |---------|------------|----------|----------|
> > | WA      | 0.42       | 0.04     | 1.00     |
> > | GA      | 0.82       | 0.48     | 0.99     |
> >
> > In this deeper network, WA and GA remain strong even in the layers that are farthest from the error signal. Transformer architectures also exhibit significant difficulty in establishing WA and GA, as illustrated in Figure 4. Applying our method to deeper Transformer models helps alleviate this issue, enabling stable alignment formation and yielding notable performance gains. Taken together, these experimental results provide strong evidence that our approach is effective even in deep network architectures.

---

> > > ### Comment · Reviewer_HVYL · 2025-11-27
> > >
> > > Thanks for the authors' rebuttal response. I will keep my score unchanged.

---

### Official Review · Reviewer_LvJQ · 2025-11-01

**Soundness:** 3
**Presentation:** 3
**Contribution:** 3
**Rating:** 6
**Confidence:** 2

**Summary:**

In this work, the authors analyzed the limitations of the current Direct Feedback Alignment (DFA) methods for fine-tuning pretrained fully connected neural networks via weight alignment (WA) and gradient alignment (GA). They further demonstrated that the proposed feedback-weight matching method, when combined with weight decay, can not only effectively mitigate over-fitting but also further reduce the network output error.

**Strengths:**

Originality: It has mitigated the limitation that DFA could only train networks from scratch, by originally proposing the feedback-weight matching method to enhance the standard DFA,,thereby enabling fine-tuning using this method.
Significance:Currently, fine-tuning still plays a role as a critical training stage for maintaining strong performance in various large models. Introducing DFA ,a method enabling efficient parallel parameter updates—into the fine-tuning process to replace back-propagation holds significance.

**Weaknesses:**

Quality:while Proposition 4.2 and Conjecture 4.3 attempt to link FWM with weight decay for error reduction, the proof relies on strong assumptions (e.g., two-layer networks, specific activation properties). The conjecture for L-layer networks remains ​​unproven​​, and the theoretical analysis does not fully account for the interplay between feedback matching and weight decay in deep networks. This lack of rigor weakens the theoretical contribution.

**Questions:**

Q1:Since the proposed feedback-weight matching method is intended to replace backpropagation,have the authors ever considered doing experiments to directly compare between the computing time of the same pretrained models but fine-tuned based on the proposed method and back-propagation?
Q2:Why the standard DFA method seems a little better than the proposed method when fine-tuning the 4-layer network from TinyImageNet in image classification tasks?

---

> ### Author Response · Authors · 2025-11-19
>
> We sincerely thank the reviewer for their careful reading of our manuscript and for providing insightful comments. We are encouraged by your acknowledgment of the relevance of our work and the clarity of our analysis. Below, we address each point raised in detail.
>
> ---
>
> **Weaknesses**
>
> We provided theoretical proofs of strong weight alignment (WA), strong gradient alignment (GA), and the effect of weight decay only for two-layer neural networks. While we have not yet mathematically proven these results for deeper architectures, we believe that our experimental results serve as empirical evidence.
>
> As shown in Figure 2, applying our method with weight decay during DFA fine-tuning significantly improves both WA and GA compared to standard DFA fine-tuning. Furthermore, the table below reports WA and GA for a deeper six-layer fully connected network trained on CIFAR-10.
>
> **Table 1. Weight Alignment and Gradient Alignment at Layer 2 in a Six-Layer Network**
> | Layer 2 | DFA_Scratch | DFA_Fine | DFA_Ours |
> |---------|------------|----------|----------|
> | WA      | 0.42       | 0.04     | 1.00     |
> | GA      | 0.82       | 0.48     | 0.99     |
>
> In the six-layer network, WA and GA values remain relatively strong even for layers far from the error signal.
>
> Transformer architectures similarly face significant difficulties in forming WA and GA, which corresponds to performance degradation as shown in Figure 4. By applying our method to deeper transformer models, alignment formation is facilitated, leading to substantial performance improvements. This demonstrates that our proposed method is effective in deeper networks.
>
> **Questions 1.**
>
> Our proposed method differs from standard DFA only in the initialization of the weights and feedback matrices; during fine-tuning, it follows the same algorithm as standard DFA. Therefore, we compare the per-layer computation time between DFA and BP. Currently, implementing layer-parallel backward updates is challenging, so we calculate DFA by splitting the gradients at each layer in the BP algorithm and replacing them with the error signal. As a result, both algorithms perform updates sequentially. By reporting per-layer computation times, we can indirectly illustrate the potential efficiency gains under parallel training.
>
> Table 2 shows the backward time for each layer in a model with six layers, each having 1,000 hidden units, trained on CIFAR-100. The total time represents the overall time required for updating all layers. In BP, computations must be performed sequentially, so the times of lower layers are included. In DFA, due to the potential for parallel computation, this is not necessary.
>
> **Table 2. DFA vs BP Layer-wise and Total Backward Time Comparison (ms), with DFA max highlighted.**
> | Type          | layers.0 | layers.1 | layers.2 | layers.3 | layers.4 | layers.5 |
> |---------------|----------|----------|----------|----------|----------|----------|
> | DFA (layer)   | 0.0132   | **0.1003**   | 0.0942   | 0.0910   | 0.0895   | 0.0811   |
> | BP (layer)    | 0.0098   | 0.0899   | 0.0927   | 0.0942   | 0.0957   | 0.0996   |
> | DFA (total)   | **0.1003**   | **0.1003**   | **0.1003**   | **0.1003**   | **0.1003**   | **0.1003**   |
> | BP (total)    | 0.4820   | 0.4722   | 0.3823   | 0.2895   | 0.1953   | 0.0996   |
>
>
> **Questions 2.**
>
> In certain conditions, $DFA_{fine}$ may achieve better results than $DFA_{our}$. However, we interpret this as an indication of the instability of standard DFA fine-tuning. $DFA_{fine}$ does not consistently perform well across all settings. For example, it does not show strong performance across all TinyImageNet variants, nor does it perform well on the 4-layer model. Even within a specific dataset, its performance is not consistent. In contrast, $DFA_{ours}$ exhibits relatively stable performance across various conditions, including different model sizes and pre-training settings. We believe this demonstrates the robustness and stability of our proposed method.

---

### Official Review · Reviewer_4qY7 · 2025-11-03

**Soundness:** 3
**Presentation:** 3
**Contribution:** 3
**Rating:** 6
**Confidence:** 2

**Summary:**

This paper investigates Direct Feedback Alignment (DFA), an alternative to back-propagation, for the task of network fine-tuning. While DFA has shown promise for training from scratch, it is known to perform unreliably in fine-tuning scenarios. The authors provide a theoretical and empirical analysis arguing that this failure stems from a fundamental mismatch between the pre-trained weights (learned via BP) and the random feedback matrices used by standard DFA, which prevents the network from achieving 'Weight Alignment' (WA). To address this, the authors propose 'Feedback-Weight Matching,' a method that first reconstructs feedback matrices from the pre-trained weights and then re-initializes the weights to align with these new matrices before applying DFA. The proposed method is evaluated on fully connected networks and Transformer models (BERT-Tiny, ViT-Tiny/Small), showing significant improvements over standard DFA fine-tuning.

**Strengths:**

1. The problem studied in this paper is interesting and potentially important for developing more efficient, BP-free learning paradigms.

2. The proposed method is well-supported by a theoretical foundation.

3. The experimental results are consistent and clear, which looks reasonable to me.

4. The presentation is generally good. The paper is easy to read.

**Weaknesses:**

1. The authors claim that DFA is 'back-propagation free,' which implies a computational efficiency advantage. However, the paper lacks any empirical comparison of running time or computational overhead (e.g., FLOPs, wall-clock time) against standard back-propagation fine-tuning. This leaves the practical benefits of the method (beyond being a BP-free algorithm) unsubstantiated.

2. The paper's scope may be somewhat narrow. The proposed method is only compared against existing DFA and standard back-propagation. Assessing its impact in the broader field of BP-free algorithms with comparisons to other methods can be a plus.

**Questions:**

Please see my comments above.

---

> ### Author Response · Authors · 2025-11-19
>
> We sincerely thank the reviewer for their thoughtful and constructive feedback on our work. We greatly appreciate the reviewer’s recognition of the importance and timeliness of the problem addressed in our paper, as well as the acknowledgement that our analysis and proposed solutions are both meaningful and well-motivated. Below, we provide detailed responses to all comments and concerns raised.
>
> ---
>
> **Weaknesses 1**
>
> Our method currently requires a small additional training cost of three epochs for weight matching. After reinitializing the weights and feedback matrices, the subsequent fine-tuning procedure is identical to standard DFA. Therefore, the training time and computational overhead during fine-tuning are the same as those of the original DFA algorithm. Theoretically, DFA passes the error of the output layer directly to each layer of the network, allowing all layers to be updated in parallel. This means that training can become faster than backpropagation, with speed improvements proportional to the number of layers. If a network has $L$ layers, the execution time can, in principle, be up to $L$ times faster than BP.
>
> However, due to implementation challenges, we were not able to conduct experiments that include inter-layer parallel updates within a single model. Most existing parallelization techniques focus on distributing the model or data across multiple GPUs, and do not consider layer-parallel updates within a single GPU. We are not certain whether we can implement this within the discussion period, but if it becomes feasible, we will include additional experiments on training time. Nonetheless, we believe that our experimental setup, which separates the gradients for each layer, adequately reflects the learning behavior of parallel updates, at least in terms of accuracy.
>
> The table below reports the backward time for each layer. We measured the backward time for a model with six layers, each having 1,000 hidden units, trained on CIFAR-100.The total time represents the overall time required for updating each layer. In backpropagation (BP), computations must be performed sequentially, so the time of the lower layers is included. In DFA, due to parallel computation, this is not necessary.
>
> **Table 1. DFA vs BP Layer-wise and Total Backward Time Comparison (ms), with DFA max highlighted.**
> | Type          | layers.0 | layers.1 | layers.2 | layers.3 | layers.4 | layers.5 |
> |---------------|----------|----------|----------|----------|----------|----------|
> | DFA (layer)   | 0.0132   | **0.1003**   | 0.0942   | 0.0910   | 0.0895   | 0.0811   |
> | BP (layer)    | 0.0098   | 0.0899   | 0.0927   | 0.0942   | 0.0957   | 0.0996   |
> | DFA (total)   | **0.1003**   | **0.1003**   | **0.1003**   | **0.1003**   | **0.1003**   | **0.1003**   |
> | BP (total)    | 0.4820   | 0.4722   | 0.3823   | 0.2895   | 0.1953   | 0.0996   |
>
> Currently, it is not possible to perform fully parallel training because the gradients of each layer are handled sequentially. However, if parallel training is feasible, the update time for a single DFA model would be determined by the layer with the longest backward time.
>
>
> **Weaknesses2**
>
> Thank you for the suggestion. In our work, our main goal was to understand and alleviate the inherent difficulties of fine-tuning when using standard DFA. For this reason, we chose not to include experiments with other BP-free algorithms, as such results would not directly support the central contribution of our study. Even if our method achieves higher accuracy than other algorithms, we do not believe that this alone demonstrates that the fine-tuning challenges specific to DFA have been resolved. That said, we agree that adding these comparisons could still provide valuable additional insights, and we will therefore proceed with the experiments.

---

### Comment · Area_Chair_Eh7E · 2025-11-27

Dear reviewers,

The authors have provided detailed responses to your reviews. I would appreciate if you could let both me and the authors know how these responses impact your assessment of the paper.

Best,

AC

---

### Author Response · Authors · 2025-12-03

We sincerely thank the reviewers for their careful and constructive feedback. Based on the comments, we have updated the manuscript accordingly.

In particular, we have added training-time and memory-usage analyses to the Training Cost section in the Appendix, providing a clearer comparison between BP and DFA under our experimental setting. All modifications are highlighted in red for ease of reference.

We appreciate the reviewers’ thoughtful suggestions, which have helped improve the clarity and completeness of the paper. Thank you again for your time and valuable input.

---

### Meta-Review · Area_Chair_gRnX · 2025-12-18

**Summary:**

**Paper Summary:** The paper introduces feedback–weight matching, an initialization strategy for fine-tuning backpropagation-pretrained models using Direct Feedback Alignment (DFA). The authors show theoretically that vanilla DFA struggles to produce strong alignment between forward and feedback weights, and that the proposed initialization effectively overcomes this limitation. They further demonstrate that the method works well with weight decay to reduce training error. Extensive experiments across a variety of vision and NLP architectures support the effectiveness of the proposed approach.

**Concerns:** Reviewers raised concerns regarding comparisons of computational cost with other BP-free algorithms, the theoretical analysis being limited to two-layer networks, the lack of experiments on large foundation models or self-attention layers, and insufficient computational cost comparisons with stronger practice-oriented baselines.

**Authors' Rebuttal:** The authors have adequately addressed most of these concerns; however, experiments on large foundation models/self-attention layers and more comprehensive computational cost comparisons with other fine-tuning methods remain missing

**Recommendation:** The AC encourages the authors to include additional experiments to further address remaining concerns. At the same time, the AC acknowledges the advantages of DFA in enabling parallel weight updates and notes that prior DFA studies have mostly focused on fully connected architectures. Given the simplicity of the method, the clear design rationale, theoretical support, and potential impact of applying DFA to fine-tuning, the AC recommends **acceptance**.

**Reviewer Concerns:**

Reviewers raised concerns regarding computational cost comparisons with other BP-free algorithms, the theoretical analysis being limited to two-layer networks, the lack of experiments on large foundation models or self-attention architectures, and comparisons with stronger, practice-oriented baselines. The authors have addressed most of these concerns; however, experiments on large foundation models/self-attention layers and comprehensive computational cost comparisons with other fine-tuning methods remain missing.

**Reviewer Scores:**

Upon detailed review, minor concerns persist. Consequently, the reviewers are expected to maintain their original scores: 6, 6, 6, and 4.

---

### Decision · Program_Chairs · 2026-01-26

Accept (Poster)